



**Estimation of the fossil-fuel component in atmospheric $CO_2$ based on**
**radiocarbon measurements at the Beromünster tall tower, Switzerland**
Tesfaye A. Berhanu[1], Sönke Szidat[2], Dominik Brunner[3], Ece Satar[1], Rudiger Schanda[1], Peter
Nyfeler[1], Michael Battaglia[2], Martin Steinbacher[3], Samuel Hammer[4] and Markus
Leuenberger[1]
[1]*Climate and Environmental Physics, Physics Institute and Oeschger Centre for Climate Change Research,*
*University of Bern, Bern, Switzerland*
[2]*Department of Chemistry and Biochemistry and Oeschger Center for Climate Change Research, University of*
*Bern, Bern, Switzerland*
[3]*Empa, Laboratory for Air Pollution/Environmental Technology, Dübendorf, Switzerland*
[4]*Institut für Umweltphysik, Universität Heidelberg, Heidelberg, Germany*
Abstract
Fossil fuel $CO_2$ ($CO_{2ff}$) is the major contributor of anthropogenic $CO_2$ in the atmosphere, and
accurate quantification is essential to better understand the carbon cycle. Since October 2012,
we have been continuously measuring the mixing ratios of CO, $CO_2$ $CH_4$ and $H_2O$ at five
different heights at the Beromünster tall tower, Switzerland. Air samples for radiocarbon
($\Delta^{14}CO_2$) analysis have also been collected from the 212.5 m sampling inlet of the tower on a
bi-weekly basis. A correction was applied for $^{14}CO_2$ emissions from nearby nuclear power
plants (NPPs), which have been simulated with the Lagrangian transport model FLEXPART-
COSMO. The $^{14}CO_2$ emissions from NPPs offset the depletion in $^{14}C$ by fossil-fuel emissions
resulting in an underestimation of the fossil-fuel component in atmospheric $CO_2$ by about 16
%. An average observed ratio ($R_{CO}$) of 13.4 ± 1.3 mmol/mol was calculated from the
enhancements in CO mixing ratios relative to the clean air reference site Jungfraujoch ($\Delta CO$)
and the radiocarbon-based fossil-fuel $CO_2$ mole fractions. This ratio is significantly higher





than both the mean anthropogenic $CO/CO_2$ emission ratios estimated for Switzerland from the
national inventory (7.8 mmol/mol for 2013), and the ratio between in-situ measured CO and
$CO_2$ enhancements at Beromünster over the Jungfraujoch background (8.3 mmol/mol).
Differences could not yet be assigned to specific processes and shortcomings of these two
methods but may originate from locally variable emission ratios as well as from non-fossil
emissions and biospheric contributions. By combining the ratio derived using the radiocarbon
measurements and the in-situ measured CO mixing ratios, a high-resolution time series of
$CO_{2ff}$ was calculated exhibiting a clear seasonality driven by seasonal variability in emissions
and vertical mixing. By subtracting the fossil-fuel component and the large-scale background,
we have determined the regional biospheric $CO_2$ component that is characterized by seasonal
variations ranging between -15 to +30 ppm. A pronounced diurnal variation was observed
during summer modulated by biospheric exchange and vertical mixing while no consistent
pattern was found during winter.
**1. Introduction**

Fossil fuel $CO_2$ ($CO_{2ff}$) is the fundamental contributor to the increase in atmospheric

$CO_2$, hence its precise quantification is crucial to better understand the global carbon budget.
One of the major uncertainties in the projections of climate change is the uncertainty in the
future carbon budget due to feedbacks between terrestrial ecosystems and climate (Heimann
and Reichstein, 2008). Information on the response of the biosphere to climate variations can
be obtained from atmospheric $CO_2$ observations, but isolating the biospheric signal in the
measured $CO_2$ mixing ratios requires an accurate quantification of the fossil fuel component.
Several methods have therefore been proposed for quantifying $CO_{2ff}$, which are based on
observations or models. A widely employed approach is to determine $CO_{2ff}$ with an
atmospheric transport model that incorporates $CO_{2ff}$ emissions from a bottom-up emission
inventory.



Emission inventories are based on statistics of the energy-use by different sectors and
the quantification of $CO_{2ff}$ emissions by accounting for the carbon content of each fuel and its
corresponding oxidation ratios (Friedlingstein et al., 2010; Le Quéré et al., 2016). When
compared to other greenhouse gases, national emission inventories for $CO_2$ are quite accurate,
but the computation of these inventories is laborious, and the quality depends on the energy
statistics and reporting methods that vary strongly between countries (Marland, 2008;
Marland et al., 2009). A recent study evaluating different energy statistics and cement
production data estimated an uncertainty of about 5 % for the global fossil-fuel emissions of
the past decade (2006 – 2015)(Le Quéré et al., 2016). At country level the uncertainties are
usually below 5 % in developed countries but often exceed 10 % in developing countries
(Ballantyne et al., 2015).
Additional uncertainties arise from the spatial and temporal disaggregation of national
annual total emissions to the grid of the atmospheric transport model. At sub-country scales
(less than 150 km), the uncertainty from bottom-up estimates can reach up to 50 % (Ciais et
al., 2010). Finally, errors in the transport model and the inability to correctly represent point
observations in the model may contribute substantially to the uncertainty of model simulated
$CO_{2ff}$ mixing ratios (Tolk et al., 2008; Peylin et al., 2011).
Radiocarbon measurements can be used to directly quantify $CO_{2ff}$ in atmospheric $CO_2$
observations. Radiocarbon is produced in the lower stratosphere during the reaction of
neutrons with nitrogen induced by cosmic rays (Currie, 2004). In addition, nuclear bomb tests
in the 1960s led to large radiocarbon input into the atmosphere which was thereafter
decreasing due to gradual uptake by the oceans and the terrestrial biosphere. Nowadays, the
decline in atmospheric $^{14}CO_2$ is mainly driven by input from $^{14}C$-free fossil fuel $CO_2$ (Levin et
al., 2010). This decline is well detectable at background sites such as Jungfraujoch,
Switzerland and Schauinsland, Germany (Levin et al., 2013). While all reservoirs exchanging



carbon with the atmosphere are relatively rich in $^{14}$C, fossil-fuels (millions of years old) are
devoid of $^{14}$C due to its radioactive decay with a half-life of 5370 years. Hence, any fossil-fuel
$CO_2$ emitted to the atmosphere will dilute the background $^{14}$C signal, the so-called Suess
effect, which can then be used to unravel recently added fossil-fuel $CO_2$ to the atmosphere
(Zondervan and Meijer, 1996; Levin et al., 2003; Gamnitzer et al., 2006; Turnbull et al., 2006;
Levin and Karstens, 2007; Turnbull et al., 2009; Turnbull et al., 2011; Lopez et al., 2013;
Turnbull et al., 2014; Turnbull et al., 2015). However, this depletion can also partially be
offset by $CO_2$ release from the biosphere which has enriched $^{14}$C/$^{12}$C ratios due to the bomb
tests as well as by direct $^{14}$C emissions from the nuclear industries (Levin et al 2010). This
technique also enables to separate between biospheric and fossil-fuel $CO_2$ components in
atmospheric $CO_2$ observations, and thus to better constrain the biospheric $CO_2$ fluxes when
coupled with inversion models (Basu et al., 2016). The uncertainty in $CO_{2ff}$ estimated by the
radiocarbon method is mainly determined by the precision in the $^{14}$C measurement, the choice
of background as well as the contribution from other sources of $^{14}$C such as nuclear power
plants (NPPs) (Turnbull et al., 2009).

Despite its importance as a fossil-fuel tracer, measurements of $^{14}$C are still sparse. The

measurements are expensive and laborious, which so far has prevented frequent sampling and
has motivated researchers to combine $^{14}$C measurements with additional tracers such as CO to
enhance spatial and temporal coverage (Gamnitzer et al., 2006; Levin and Karstens, 2007;
Vogel et al., 2010; Lopez et al., 2013; Turnbull et al., 2014; Turnbull et al., 2015). The CO-
method relies on using high frequency CO measurements and regular calibration of the
temporally changing $\Delta CO:\Delta CO_{2ff}$ ratios based on weekly or bi-weekly $^{14}$C measurements.
Despite its advantage of providing a proxy for continuous $CO_{2ff}$ data, the method introduces
additional uncertainties due to diurnal and seasonal variability in the CO sink, and the
presence of multiple non-fossil CO sources such as oxidation of hydrocarbons or wood and




biofuel combustion (Gamnitzer et al., 2006). Spatial variations in the $\Delta CO:\Delta CO_2$ ratio across
Europe due to different source compositions and environmental regulations, which affects the
measured ratios due to changes in air mass origin (Oney et al., In review) are the main reason
for the temporally changing $\Delta CO:\Delta CO_{2ff}$ ratio for a given measurement site.

In Switzerland, $CO_2$ contributes about 82 % of the total greenhouse gas emissions

according to the Swiss national emission inventory for 2013, and fossil-fuel combustion from
the energy sector contributes more than 80 % of the total $CO_2$ emission (FOEN, 2015b). In
order to validate such bottom-up estimates, independent techniques based on atmospheric
measurements are desirable. In addition, as mentioned above, the biospheric $CO_2$ signals can
only be estimated with a good knowledge of $CO_{2ff}$. In this study, we present and discuss
$^{14}CO_2$ measurements conducted bi-weekly between 2013 and 2015 at the Beromünster tall
tower in Switzerland. From these samples in combination with background CO, $CO_2$ and
$^{14}CO_2$ measurements at the high-altitude remote location Jungfraujoch, Switzerland, $\Delta CO$ to
$\Delta CO_{2ff}$ ratios ($R_{CO}$) are derived. These ratios are then combined with the in-situ measured
$\Delta CO$ mixing ratios to estimate a high-resolution time series of atmospheric $CO_{2ff}$ mixing
ratios, and by difference, of the biospheric $CO_2$ component. The influence of $^{14}C$ emissions
from nearby NPPs and correction strategies are also discussed.
**2. Methods**
**2.1. Site description and continuous measurement of CO and $CO_2$**

A detailed description of the Beromünster tall tower measurement system as well as a

characterization of the site with respect to local meteorological conditions, seasonal and
diurnal variations of greenhouse gases, and regional representativeness can be obtained from
previous publications (Oney et al., 2015; Berhanu et al., 2016; Satar et al., 2016). In brief, the
tower is located near the southern border of the Swiss Plateau, the comparatively flat part of
Switzerland between the Alps in the south and the Jura mountains in the northwest (47° 11′



23″ N, 8° 10' 32″ E, 797 m a.s.l.), which is characterized by intense agriculture and rather
high population density (Fig. 1). The tower is 217.5 m tall with access to five sampling
heights (12.5 m, 44.6 m, 71.5 m, 131.6 m, 212.5 m) for measuring CO, $CO_2$, $CH_4$ and $H_2O$
using Cavity Ring Down Spectroscopy (CRDS) (Picarro Inc., G-2401). By sequentially
switching from the highest to the lowest level, mixing ratios of these trace gases were
recorded continuously for three minutes per height, but only the last 60 seconds were retained
for data analysis. The calibration procedure for ambient air includes measurements of
reference gases with high and low mixing ratios traceable to international standards (WMO-
X2007 for $CO_2$ and WMO-X2004 for CO and $CH_4$), as well as target gas and more frequent
working gas determinations to ensure the quality of the measurement system. From two years
of data a long-term reproducibility of 2.79 ppb, 0.05 ppm, and 0.29 ppb for CO, $CO_2$ and
$CH_4$, respectively was determined for this system (Berhanu et al., 2016).
**2.2. Sampling and $CO_2$ extraction for isotope analysis**
Air samples for $^{14}CO_2$ analysis were collected every second week from the highest
inlet usually between 9:00 to 13:00 UTC. During each sampling event, three samples were
collected over a 15-minute interval in 100 L PE-AL-PE bags (TESSERAUX, Germany) from
the flush pump exhaust line of the 212.5 m sampling inlet, which has a flow rate of about 9 L
$min^{-1}$ at ambient conditions. The sampling interval was chosen to ensure radiocarbon sample
collection in parallel with the continuous CO and $CO_2$ measurements by the CRDS analyzer
at the highest level. Each bag was filled at ambient air pressure for 6 to 8 minutes and a total
air volume of 50 to 70 L (at STP) was collected.
$CO_2$ extraction was conducted cryogenically in the laboratory at the University of
Bern usually the day after the sample collection. During the extraction step, the air sample
was first pumped through a stainless steel water trap (-75 °C), which was filled with glass
beads (Rashig rings, 5 mm, Germany). A flow controller (Analyt-MTC, Aalborg, USA) with





flow totalizer tool was attached to this trap to maintain a constant flow of air (1.2 L min$^{-1}$)
towards the second trap (trap 2), a spiral-shaped stainless steel tube (1/4") filled with glass
beads (~ 2 mm) and immersed in liquid nitrogen to freeze out $CO_2$. When the flow ceased,
trap 2 was isolated from the line and evacuated to remove gases which are non-condensable at
this temperature. Then, trap 2 was warmed to room temperature, and eventually immersed in
slush at -75 °C to freeze out any remaining water. Finally, the extracted $CO_2$ was expanded
and collected in a 50 mL glass flask immersed in liquid nitrogen.

Sample extraction efficiency was calculated by comparing the amount of the

cryogenically extracted $CO_2$ with the $CO_2$ measured in-situ by the CRDS analyzer during the
time of sampling. The amount of $CO_2$ extracted is determined first by transferring the
extracted $CO_2$ cryogenically to a vacuum line of predetermined volume. Then, based on the
pressure reading of the expanded gas, and the total volume of air collected determined by the
mass flow controller with a totalizer function attached to trap 1, $CO_2$ mixing ratios were
calculated.

At the end of 2014 we noticed that there was a leakage from the sampling line exhaust

pumps, which resulted in unrealistically high $CO_2$ mixing ratios (usually more than 500 ppm).
Therefore, we repalced all the exhaust pumps and the leakage problem was solved. Seven
samples, which were suspected to be contaminated due to this issue, were consequently
excluded. The sample extraction efficiency since then has usually been better than 99 %. We
also made a blank test to check the presence of any leaks or contamination during sample
processing but did not observe any of these issues.
2.3. Measurement of $\delta^{13}$C, $\delta^{18}$O and $\Delta^{14}$C

Prior to radiocarbon measurement, the extracted $CO_2$ was analyzed for the stable

isotopes $\delta^{13}$C and $\delta^{18}$O using the Isotope Ratio Mass Spectrometer (Finnigan MAT 250) at the
Climate and Environmental Physics Division of University of Bern, which has an accuracy





and precision of better than 0.1 ‰ for both $\delta^{13}C$ and $\delta^{18}O$ (Leuenberger et al., 2003). $^{14}C$
analysis of the extracted $CO_2$ was performed with an accelerator mass spectrometer (AMS)
MICADAS (MIni CArbon DAting System) at the Laboratory for the Analysis of Radiocarbon
(LARA) at the Department of Chemistry and Biochemistry of the University of Bern (Szidat
et al., 2014). An automated graphitization equipment (AGE) was used to prepare solid target
gas (Nemec et al., 2010) from the extracted $CO_2$ stored in 50 mL glass flasks. A measurement
series consisted of up to 15 air samples converted to 30 solid graphite targets (duplicates),
together with four and three targets from $CO_2$ produced by combustion of the NIST standard
oxalic acid II (SRM 4990C) and fossil $CO_2$ (Carbagas, Gümligen), respectively, which were
used for the blank subtraction, standard normalization, and correction for isotopic
fractionations. Data reduction was performed using the BATS program (Wacker et al., 2010).
As $^{14}C/^{12}C$ from Beromünster was measured at the LARA laboratory in Bern, whereas the
corresponding background samples from Jungfraujoch were analyzed at the Institute of
Environmental Physics, Heidelberg University, the datasets needed to be adjusted to each
other. A recent interlaboratory compatibility test estimated a small bias of $2.1 \pm 0.5$ ‰
(Hammer et al., 2016) between the two institutes, which was subsequently subtracted from the
$^{14}C$ measurements of the Beromünster samples.
2.4. Determination of the fossil fuel $CO_2$ component
2.4.1. The $\Delta^{14}C$ technique
For the determination of the $CO_{2ff}$ component we followed the method developed by
Levin and co-workers (Levin et al., 2003; Levin and Karstens, 2007). The measured $CO_2$ is
assumed to be composed of three major components: the free troposphere background
($CO_{2bg}$), the regional biospheric component ($CO_{2bio}$) comprising photosynthesis and
respiration components, and the fossil-fuel component ($CO_{2ff}$):
$$CO_{2meas} = CO_{2bg} + CO_{2bio} + CO_{2ff} \qquad (1)$$





Each of these components has a specific $\Delta^{14}C$ value described as $\Delta^{14}C_{meas}$, $\Delta^{14}C_{bg}$, $\Delta^{14}C_{bio}$ and
$\Delta^{14}C_{ff}$. In analogy to Eq. (1), a mass balance equation can also be formulated for $^{14}C$ as:
$\quad CO_{2meas} \, (\Delta^{14}C_{meas} + 1000\ \text{‰}) = CO_{2bg} \, (\Delta^{14}C_{bg} + 1000\ \text{‰}) + CO_{2bio} \, (\Delta^{14}C_{bio} + 1000$
$\quad \text{‰}) + CO_{2ff} \, (\Delta^{14}C_{ff} + 1000\ \text{‰}) \hfill (2)$
Note that non-fossil fuel components such as biofuels are incorporated into the biospheric
component in Eq. (1). The fossil-fuel term in Eq. (2) is zero as fossil fuels are devoid of
radiocarbon ($\Delta^{14}C_{ff}$ = -1000 ‰). Replacing the biospheric $CO_2$ component in Eq. (1) by a
formulation derived from Eq. (2), the fossil fuel $CO_2$ component is derived as:
$$CO_{2ff} = \frac{CO_{2bg}\left(\Delta^{14}C_{bg} - \Delta^{14}C_{bio}\right) - CO_{2meas}\left(\Delta^{14}C_{meas} - \Delta^{14}C_{bio}\right)}{\Delta^{14}C_{bio} + 1000\text{‰}} \qquad (3)$$
Equation (3) can be further simplified by assuming that $\Delta^{14}C_{bio}$ is equal to $\Delta^{14}C_{bg}$ (Levin et al.,
2003) as:
$$CO_{2ff} = \frac{CO_{2meas}\left(\Delta^{14}C_{bg} - \Delta^{14}C_{meas}\right)}{\Delta^{14}C_{bg} + 1000\text{‰}} \qquad (4)$$
Hence, the fossil fuel $CO_2$ component can be determined using the $CO_{2meas}$ and $\Delta^{14}C_{meas}$
values measured at the site as well as $\Delta^{14}C_{bg}$ obtained from the Jungfraujoch mountain
background site in the Swiss Alps.
However, the $CO_{2ff}$ determined using Eq. (4) incorporates a small bias due to the non-
negligible disequilibrium contribution of heterotrophic respiration while the contribution from
autotrophic respiration can be approximated by $\Delta^{14}C_{bg}$. Turnbull et al. (2006) showed that this
effect will lead to an underestimation of $CO_{2ff}$ on average by 0.2 ppm in winter and 0.5 ppm
in summer, respectively, estimated for the northern hemisphere using a mean terrestrial
carbon residence time of 10 years. To account for this bias, a harmonic function varying
seasonally between these values was added to the derived $CO_{2ff}$ values. However, variation of
respiration fluxes on shorter timescales cannot be accounted for by this simple correction.



### 2.4.2. Simulation of $^{14}CO_2$ from nuclear power plants


Radiocarbon is produced by nuclear reactions in NPPs and primarily emitted in the
form of $^{14}CO_2$ (Yim and Caron, 2006), except for Pressurized Water Reactors (PWR), which
release $^{14}C$ mainly in the form of $^{14}CH_4$. Previous studies have shown that such emissions can
lead to large-scale gradients in atmospheric $\Delta^{14}C$ activity and offset the depletion from fossil-
fuel emissions (Graven and Gruber, 2011). At Heidelberg in Germany, an offset of 25 % and
10 % of the fossil-fuel signal was observed during summer and winter, respectively, due to
emissions from a nearby plant (Levin et al., 2003). Similarly, Vogel et al. (2013) determined
the influence of NPPs for a measurement site in Canada, and estimated that about 56 % of the
total $CO_{2ff}$ component was masked by the contribution from NPPs. In Switzerland, there are
five NPPs and the closest plant is located about 30 km to the northwest of Beromünster (Fig.
1). Furthermore, air masses arriving at Beromünster are frequently advected from France,
which is the largest producer of nuclear power in Europe.
To estimate the influence of NPPs on $\Delta^{14}C$ at Beromünster, we used FLEXPART-
COSMO backward Lagrangian particle dispersion simulations (Henne et al., 2016).
FLEXPART-COSMO was driven by hourly operational analyses of the non-hydrostatic
numerical weather prediction model COSMO provided by the Swiss weather service
MeteoSwiss at approximately 7 x 7 km$^2$ resolution for a domain covering large parts of
Western Europe. For each 3-hour measurement interval during the three-year period, a source
sensitivity map (footprint) was calculated by tracing the paths of 50'000 particles released
from Beromünster at 212 m above ground over 4 days backward in time. The source
sensitivities were then multiplied with the $^{14}CO_2$ emissions of all NPPs within the model
domain. Thereby, the emission of a given NPP was distributed over the area of the model grid
cell containing the NPP. Source sensitivities were calculated for three different vertical layers
(0-50 m, 50-200 m, 200-500 m). Since the height of ventilation chimneys of the Swiss NPPs



is between 99 m and 120 m, only the sensitivity of the middle layer was selected here as it
corresponds best to the effective release height.

The release of $^{14}$C both in inorganic ($CO_2$) and organic form ($CH_4$) is routinely

measured at all Swiss NPPs. The corresponding data have been kindly provided by the Swiss
Federal Nuclear Safety Inspectorate (ENSI) and the Berner Kraftwerke (BKW) operating the
NPP Mühleberg at temporal resolutions ranging from annual (Benznau 1 & 2), to monthly
(Leibstadt, Gösgen), and bi-weekly (Mühleberg), and we assumed constant emissions over the
corresponding periods. For Beznau 1, the emissions of 2015 were distributed over the first 3
months of the year due to the shut-down of the plant in March 2015. The largest sources of
$^{14}CO_2$ in Switzerland are the two Boiling Water Reactors (BWP) Mühleberg and Leibstadt
(Loosli and Oeschger, 1989). Beznau 1 & 2 and Gösgen are PWRs emitting about one order
of magnitude less $^{14}CO_2$. For NPPs outside Switzerland, the emissions were estimated from
energy production data reported to the International Atomic Energy Agency (IAEA) and NPP
type-specific emission factors following Graven and Gruber (2011). The enhancement in $\Delta^{14}$C
caused by nuclear emissions at Beromünster ($\delta\Delta_{nuc}$) was then computed according to Graven
and Gruber (2011) as:
$$\delta\Delta_{nuc} = \frac{\delta A_{nuc} \times 1000‰}{R_S(C_R + \delta C_{ff})}$$   (5)
where $\delta A_{nuc}$ and $\delta C_{ff}$ are the enhancements in $^{14}CO_2$ and $CO_2$ relative to a reference site with
a background $CO_2$ mixing ratio $C_R$, respectively. $R_S$ represents the modern day $^{14}C/^{12}C$ ratio
of $1.176 \times 10^{-12}$.
**2.4.3. Calculation of $R_{CO}$, $\Delta CO/\Delta CO_2$ and high resolution $CO_{2ff}$**

A $\Delta CO$ to $\Delta CO_{2ff}$ ratio ($R_{CO}$) was calculated as the slope of the geometric mean

regression (model II), with $\Delta CO$ being the corresponding CO enhancement over a background
measured at Jungfraujoch, and the $CO_{2ff}$ values determined above. The CO measurements at





Jungfraujoch were conducted using a CRDS analyzer (Picarro Inc., G-2401) with a
measurement precision of ±1 ppb for 10-minute aggregates (Zellweger et al., 2012).
As CO is usually co-emitted with $CO_2$ during incomplete combustion of fossil and
other fuels, we have also computed a tracer ratio designated as $\Delta CO/\Delta CO_2$ from the
enhancements in the in-situ measured CO and $CO_2$ mixing ratios over the Jungfraujoch
background  (Oney et al., 2016, In review). $CO_{2bg}$ values were obtained by applying the
robust extraction of baseline signal (REBS) statistical method to the continuous $CO_2$
measurements at the high altitude site Jungfraujoch (Schibig et al., 2016) with a band width of
60 days. Note that while $R_{CO}$ strictly refers to the ratio of $\Delta CO$ to fossil fuel $CO_2$ emissions,
the $\Delta CO/\Delta CO_2$ ratio can be influenced by biospheric contribution as well as $CO_2$ emissions
from non-fossil sources such as biofuels and biomass burning.
In order to construct the high resolution $CO_{2ff}$ time series, we combined the in-situ
measured CO enhancements at the Beromünster tower with the radiocarbon-derived ratios
$R_{CO}$, and estimated $CO_{2ff}^{CO}$ for the three-year dataset as:
$$CO_{2ff}^{CO} = \frac{CO_{obs} - CO_{bg}}{R_{CO}} \qquad (6)$$
where $CO_{obs}$ is the hourly averaged CO measurements at the tower. $CO_{bg}$ is the background
CO values derived from measurements at the High Alpine Research Station Jungfraujoch,
estimated in the same way as $CO_{2bg}$ by applying the REBS statistical method (Ruckstuhl et
al., 2012) with a bandwidth of 60 days to eliminate the influence of short-term local
variability occasionally observed at Jungfraujoch.
**3. Results and Discussions**
**3.1. $\Delta^{14}CO_2$ and $CO_{2ff}$**
Figure 2a shows the in-situ measured hourly mean $CO_2$ dry air mole fractions at
Beromünster (black) from the 212.5 m sample inlet matching at hours when air samples were



collected for radiocarbon analysis and the corresponding background $CO_2$ at Jungfraujoch
(blue). During the measurement period, we have recorded $CO_2$ mixing ratios between 389
ppm and 417 ppm. Spikes of $CO_2$ were observed mainly during winter, associated with weak
vertical mixing and enhanced anthropogenic emissions while lower $CO_2$ mixing ratios were
recorded during summer due to strong vertical mixing and photosynthetic uptake (Berhanu et
al., 2016; Satar et al., 2016).

Isotopic analysis of the air samples yielded $\Delta^{14}C_{meas}$ between -12.3 ‰ and +22.8 ‰,

with no clear seasonal trend, after correction for the model-simulated contribution from NPPs
(Fig. 2b). Based on the simulations described in section 2.4.2, we have calculated a mean
enhancement in $\Delta^{14}C$ of +1.6 ‰ and a maximum of +8.4 ‰ due to NPPs. This agrees
qualitatively with the coarse resolution simulations of Graven and Gruber (2011), which
suggest a mean enhancement of +1.4 ‰ to +2.8 ‰ over this region (Graven and Gruber,
2011). While about 70 % of this contribution is due to Swiss NPPs, the remaining
contribution is of foreign origin. About 75 % of the contribution from the Swiss NPPs is due
to Mühleberg, which is located west of Beromünster and hence frequently upstream of the
site, due to the prevailing westerly winds (Oney et al., 2015). Note that each data point
represents a mean value of the triplicate samples collected consecutively with a standard error
of 2 ‰ among triplicates. During this period, the background $\Delta^{14}C$ values measured at
Jungfraujoch varied between 15 ‰ and 28 ‰. Regional depletions in $\Delta^{14}C$ due to fossil-fuel
emissions, i.e. differences between Beromünster and the clean air reference site Jungfraujoch,
were in the range of -0.7 ‰ to -29.9 ‰ with a mean value of -9.9 ‰.

Figure 2c shows the corresponding $CO_{2ff}$ determined after correcting for radiocarbon

emissions from NPPs. The typical uncertainty in $CO_{2ff}$ is 1.1 ppm calculated from a mean
$\Delta^{14}C$ measurement uncertainty of 2.0 ‰ in both the sample and the background values. A
mean fossil-fuel $CO_2$ contribution of 4.3 ppm was calculated from these samples. Few cases,



notably the sample from 27 March 2014, showed a higher $CO_{2ff}$ and a strong depletion in
$\Delta^{14}C_{meas}$, consistent with the high $CO_2$ mixing ratio shown in the top panel. This can be due to
a strong local fossil-fuel contribution or a polluted air mass transported from other regions of
Europe coinciding with the grab samplings. As this event occurred during a period with
moderate temperatures (mean temperature of 6.8 °C measured at the highest level of the
Beromünster tower between March and May), strong fossil fuel $CO_2$ emissions due to heating
are not expected. The FLEXPART-COSMO transport simulations for this event suggest an air
mass origin from southeastern Europe. Periods with winds from the east, colloquially known
as Bise, are well known to be associated with very stable boundary layers and
correspondingly strong accumulation of air pollutants during the cold months of the year
between autumn and spring. Air masses reaching Beromünster from Eastern Europe have
recently been reported to contain unusually high levels of CO during late winter and early
spring periods, coinciding with this sampling period (Oney et al., 2016, In review).
By subtracting the background and fossil-fuel $CO_2$ contributions from the measured
mixing ratios, $CO_{2bio}$ values were also determined ranging between +11.2 ppm and -12.4 ppm
(Figure 2d). Even if there is no clear seasonal trend, the lowest $CO_{2bio}$ values were recorded
during summer implying net photosynthetic $CO_2$ uptake while most of the values in winter are
positive or close to zero due to respiration. Two of the samples in June and July 2015 showed
a rather large positive $CO_{2bio}$ contribution, in contrast to the expected summertime minimum.
Reasons for such high values can be biomass harvesting or enhanced respiration by plants and
soil, associated with warmer temperature which will lead to enhanced $CO_2$ emissions (Oney et
al., 2016, In review).
**3.2. $R_{CO}$ values from radiocarbon measurements**
From the simultaneous CO and radiocarbon measurements, we calculated an $R_{CO}$ of
13.4 ± 1.3 mmol CO/mol $CO_2$ with a correlation coefficient ($r^2$) of 0.7, and a median value of




11.2 mmol CO/mol $CO_2$. If we split the data seasonally, $R_{CO}$ values of 12.5 ± 3.3 mmol
CO/mol $CO_2$ and 14.1 ± 4.0 mmol CO/mol $CO_2$ were obtained during winter and summer,
respectively. The slightly lower $R_{CO}$ during winter is due to larger $CO_{2ff}$ share during this
period from domestic heating. Our estimate is well within the range of values from previous
studies (10-15 mmol/mol) observed at other sites in Europe and North America (Gamnitzer et
al., 2006; Vogel et al., 2010; Turnbull et al., 2011). To test the sensitivity of this ratio to the
selection of background site, we additionally calculated $R_{CO}$ using background values
estimated with the REBS method from the in-situ CO measurements at Beromünster instead
of Jungfraujoch. The value obtained in this way (12.7 ± 1.2, $r^2 = 0.6$) is only slightly lower
than the value obtained using Jungfraujoch as background site. Considering the persistent
decrease in CO emissions (Zellweger et al., 2009) in response to the European emission
legislation, our estimated $R_{CO}$ is surprisingly high.

### 359    3.3. ΔCO/ΔCO$_2$ from continuous measurements

Figure 3 shows the seasonally resolved ΔCO to ΔCO$_2$ correlations derived from in-situ

measured CO and $CO_2$ enhancements over the background observed at Jungfraujoch, and we
have estimated a tracer ratio of 8.3 ± 0.1 mmol/mol ($r^2 = 0.5$) for the entire measurement
period. From measurements during winter, when the two species are most strongly correlated,
a ΔCO/ΔCO$_2$ ratio of 7.3 ± 0.1 mmol/mol ($r^2 = 0.9$) is obtained, while barely any correlation
is observed in summer and weak correlations ($r^2 < 0.4$) during spring and autumn. Recently,
Oney et al. (2016) reported a higher wintertime ratio of 8.3 mmol/mol for the same
combination of measurements at Beromünster and Jungfraujoch but for a different time
period. If we consider only winter 2013 as in their data, we obtain essentially the same value,
while much lower ratios of 6.5 mmol/mol and 6.4 mmol/mol were calculated for 2014 and
2015, respectively. The higher ratios in winter 2013 are likely related to the unusually cold
conditions and extended periods of air mass transport from Eastern Europe. Note that these





enhancement ratios also include emissions from non-fossil sources such as biofuels and
biomass burning in contrast to $R_{CO}$. The national inventory attributes about 15 % of total $CO_2$
emissions in 2014 to non-fossil fuel sources (FOEN, 2015b). If we correct for these sources
assuming a constant contribution throughout the year, the winter time $\Delta CO/\Delta CO_2$ ratio for the
three year data becomes 8.7 mmol/mol. This ratio is roughly consistent with the
anthropogenic CO to $CO_2$ emission ratio of 7.8 mmol/mol calculated from Switzerland's
greenhouse gas inventory report for 2013 (FOEN, 2015b, a). The slightly higher value points
towards an underestimation of the CO emissions by the inventory.

This wintertime $\Delta CO/\Delta CO_2$ ratio of 8.7 mmol/mol is still about 30 % lower than the

$R_{CO}$ estimate for the same period (12.5 mmol/mol) shown as a black line in Fig. 3. This
suggests that despite the strong correlation between $\Delta CO$ and $\Delta CO_2$ in winter the regional
$CO_2$ enhancements are not only caused by anthropogenic emissions but include a significant
biospheric $CO_2$ component. This also implies that the observed correlation is not only due to
spatially and temporally correlated sources but is caused to a large extent by meteorological
variability associated with more or less accumulation of trace gases in the boundary layer
irrespective of their origin. This interpretation is supported by the fact that a strong correlation
($r^2 > 0.7$) was also observed between CO and $CH_4$ during winter at the same tower site (Satar
et al., 2016) despite their sources being vastly distinct. In Switzerland about 80 % of $CH_4$
emissions are from agriculture (mainly from ruminants) while more than 85 % of CO
emissions are from the transport sector and residential heating (FOEN, 2015a).

As a consequence, the true ratio of CO to anthropogenic $CO_2$ may be significantly

larger than the observed $\Delta CO/\Delta CO_2$ ratio of 8.7 mmol/mol, which in turn would imply that
the molar ratio of the Swiss emission inventory of 7.8 mmol/mol is too small.
**3.4. High resolution time series of $CO_{2ff}$ and $CO_{2bio}$**





Figure 4 shows the hourly mean CO mixing ratios at Jungfraujoch and Beromünster
between 2013 and 2015. CO mixing ratios as high as 480 ppb were recorded at Beromünster
while generally lower CO values were recorded at the more remote site Jungfraujoch. A
pronounced seasonality in CO can be observed at Beromünster with higher values in winter
and lower values during summer due to stronger vertical mixing and chemical depletion of
CO by OH (Satar et al., 2016). The hourly mean $CO_{2ff}$ time series calculated using these
continuous CO measurements and the seasonally resolved $R_{CO}$ values derived using the
radiocarbon measurements are displayed in Fig. 4c. A seasonal trend in the calculated $CO_{2ff}$ is
observed with frequent spikes of $CO_{2ff}$ during winter while summer values show less
variability. We calculated a monthly mean amplitude (peak-to-trough) of 6.3 ppm with a
maximum in February and a minimum in July. During the measurement period, we have
observed $CO_{2ff}$ mixing ratios ranging up to 27 ppm coinciding with cold periods and likely
from enhanced anthropogenic emissions due to heating. Instances of slightly negative $CO_{2ff}$
contributions, which occurred during less than 5 % of the time, were associated with negative
enhancements in CO (i.e. $\Delta CO < 0$). This could be simply due to an overestimation of
background values by the REBS function during these periods.
Figure 5a shows the hourly averaged residual $CO_{2bio}$ values which exhibit a clear
seasonal cycle but also a considerable scatter in all seasons ranging from -13 ppm to +30
ppm. During winter, most values were close to zero or positive, implying a dominance of
respiration fluxes. In summer, conversely, pronounced negative and positive excursions were
observed mostly due to the diurnal cycle in net $CO_2$ fluxes, which are dominated by
photosynthetic uptake during daytime and respiration at night. Another factor contributing to
such variations may be the application of a constant emission ratio neglecting any diurnal
variability (Vogel et al., 2010).





It should also be noted that any non-fossil fuel $CO_2$ sources such as emissions from
biofuels would be incorporated into the $CO_{2bio}$ term since $CO_{2ff}$ in Eq. (1) represents the
fossil-fuel sources only, adding more variability to the data set. In order to reduce the
influence of these diurnal factors, we have looked into afternoon $CO_{2bio}$ values (12:00 - 15:00
UTC), when the $CO_2$ mixing ratios along the tower are uniform (Satar et al., 2016) and $R_{CO}$
variability is minimal. Similar to the seasonal pattern in Fig. 5a, a clear seasonal cycle in
biospheric $CO_2$ can be observed (Fig. 5b) in agreement with biospheric exchange, but both
positive and negative extremes are less frequently observed (-12 ppm to +22 ppm).
The variation in $CO_{2bio}$ during afternoon (12:00 – 15:00 UTC) was recently estimated
at this site to a range of -20 ppm to +20 ppm by combining observations and model
simulations for the year 2013 (Oney et al., 2016, In review). Our estimates are more positive
when compared to their study, due to the higher $R_{CO}$ which results in lower $CO_{2ff}$ and
correspondingly higher $CO_{2bio}$ values.
Biospheric $CO_2$ shows a seasonally dependent diurnal variation as shown in Fig. 6.
During winter (Dec - Feb), the biospheric $CO_2$ component remains consistently positive (+2
to +5 ppm) throughout the day, implying net respiration fluxes. In summer, a clear feature
with increasing $CO_{2bio}$ values during the night peaking between 07:00 and 08:00 UTC (i.e.
between 08:00 and 09:00 local time) can be observed. This buildup during the night can be
explained by $CO_2$ from respiration fluxes accumulating in the stable and shallow nocturnal
boundary layer. Then, after sunrise, the early morning $CO_{2bio}$ peak starts to gradually decrease
due to a combination of onset of photosynthesis and enhanced vertical mixing due to the
growth of the boundary layer. At Beromünster, a decrease in $CO_2$ mixing ratios from both
processes is visible more or less at the same time at the 212.5 m height level, while at the
lowest inlet level (12.5 m) the photosynthetic uptake signal is observed about an hour earlier
(Satar et al., 2016). Between 12:00 and 15:00 UTC, when the daytime convective boundary



layer is fully established, the biospheric $CO_2$ continues to become more negative implying net
photosynthetic uptake, which eventually stabilizes for 3 - 5 hours until nighttime $CO_{2bio}$
accumulation starts.

**4. Conclusions**

From continuous measurements of CO and $CO_2$ and bi-weekly radiocarbon samples at
the Beromünster tall tower, we have estimated a $\Delta CO$ to $\Delta CO_{2ff}$ ratio ($R_{CO}$) which was
subsequently used to construct a 3-years long high-resolution $CO_{2ff}$ time series. We have
corrected the ratio for an offset of about 16 % caused by $^{14}C$ emissions from nearby NPPs.
This bias was calculated by comparing the simulated mean enhancement in $\Delta^{14}C$ (1.6 ‰) due
to NPPs with the measured mean depletion in $\Delta^{14}C$ due to fossil fuel $CO_2$ (9.9 ‰). The
radiocarbon-based $R_{CO}$ derived in this study during winter is significantly higher than the
$CO:CO_2$ enhancement ratios estimated from continuous CO and $CO_2$ measurements during
the same period, suggesting a significant biospheric contribution to regional $CO_2$
enhancements during this period.
The obtained $CO_{2ff}$ time series shows a clear seasonality with frequent spikes during
winter associated with enhanced anthropogenic emissions and weak vertical mixing while
summer values are mostly stable.
By subtracting the estimated $CO_{2ff}$ and $CO_{2bg}$ from $CO_{2meas}$, we have also calculated
the biospheric $CO_2$ component, which ranges between -15 ppm and +30 ppm. Considering
only afternoon data (12:00 – 15:00 UTC) when the convective boundary layer is fully
established, $CO_{2bio}$ showed its minimum in summer coinciding with net photosynthetic uptake
but still with frequent positive excursions possibly due to biomass burning or enhanced
soil/plants exhalation. During winter, $CO_{2bio}$ becomes nearly zero or positive, implying
respiration fluxes.



A pronounced diurnal variation in $CO_{2bio}$ was observed during summer modulated by
vertical mixing and biospheric exchange while this variation disappears during winter.
However, the variation in $CO_{2bio}$ may also be influenced by the uncertainty of the $CO_{2ff}$
estimate especially due to applying a constant emission ratio while calculating $CO_{2ff}$. Hence,
it will be important in the future to include seasonally and diurnally resolved $R_{CO}$ values from
high-frequency radiocarbon measurements to better estimate $CO_{2ff}$. Additionally, including
independent tracers such as Atmospheric Potential Oxygen (APO) estimates based on
concurrent $CO_2$ and $O_2$ measurements will be very useful to validate fossil-fuel emission
estimates from the radiocarbon method. This technique is also advantageous as the fossil fuel
$CO_2$ estimate is unaltered by contribution from NPPs as well as it accounts for the
contribution from biofuels.

**Acknowledgements**
This project was funded by the Swiss National Science Foundation through the Sinergia
project CarboCount CH (CRSII2 136273). We are also grateful to the ICOS-Switzerland and
the International Foundation High Alpine Research Stations Jungfraujoch and Gornergrat.
The LARA laboratory would like to thank René Fischer for the production of large $CO_2$
amounts by combustion of the NIST standard oxalic acid II, and Dejan Husrefovic for the
evaluation of the sample transfer line. Finally, we would like to thank Heather Graven and
Nicolas Gruber for helpful input regarding radiocarbon emissions from NPPs and the Swiss
Federal Nuclear Safety Inspectorate (ENSI) and the Berner Kraftwerke (BKW) for fruitful
discussions and providing radiocarbon emission data.







List of Tables and Figures
Table 1. Ratios ($R_{CO}$) determined using radiocarbon measurements after correcting for
influence from NPPs and applying model II regression, and ratios derived from continuous
CO and $CO_2$ measurements by the CRDS analyzer as enhancements ($\Delta CO:\Delta CO_2$) using
Jungfraujoch background measurements. $R_{CO}$ values are given in mmol/mol with standard
uncertainties of the slope and $r^2$ values in brackets and n represents the number of samples for
the radiocarbon method. Note that according to the Swiss emission inventory report for
greenhouse gas emissions in 2013, the annual anthropogenic $CO/CO_2$ emission ratio for the
national estimate is 7.8 mmol/mol.

|  | $R_{CO}$ ($\Delta CO:\Delta CO_{2ff}$) (radiocarbon) | Number of samples ($n$) | $\Delta CO:\Delta CO_2$ (CRDS) |
|---|---|---|---|
| Winter (Dec-Feb) | 12.5 ± 3.3 (0.6) | 8 | 7.3 (0.9) |
| Summer (Jun-Aug) | 14.1 ± 4.0 (0.3) | 14 | 13.4 (0.02) |
| All data | 13.4 ± 1.3 (0.6) | 45 | 8.3 (0.5) |








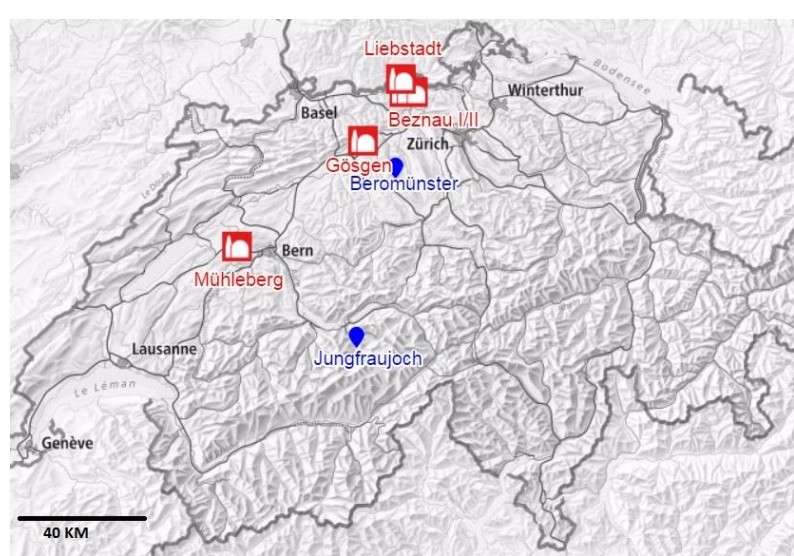


Figure 1. The geographical map of Beromünster and Jungfraujoch measurement sites (blue) as
well as the five NPPs in Switzerland (red).












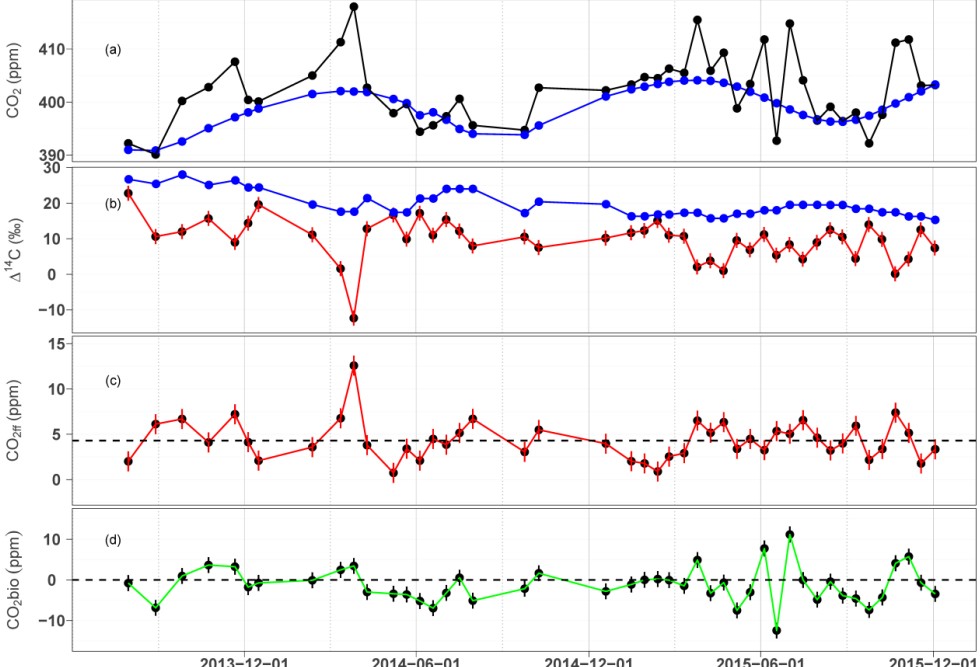


Figure 2. $CO_2$ mixing ratios (hourly averages) at Beromünster (black) from the sample inlet at

212.5 m and from background measurements at Jungfraujoch (blue) filtered using the REBS

function for periods when $^{14}C$ sampling was conducted (a), $\Delta^{14}C$ determined from the bi-

weekly point samplings at the site (red) and from 14-days integrated samplings at

Jungfraujoch (blue) (b), $CO_{2ff}$ determined during this period applying Eq. (4) with a mean

$CO_{2ff}$ value of 4.3 ppm (dashed line) (c), and the biospheric $CO_2$ determined by simple

subtraction of $CO_{2bg}$ and $CO_{2ff}$ from the $CO_{2meas}$ (d). Error bars in (b) and (c) indicate the

mean uncertainty in $\Delta^{14}C$ measurement ($\pm$ 2.1 ‰) and calculated $CO_{2ff}$ ($\pm$ 1.1 ppm), averaged

for the triplicate samples while error bar in (d) is obtained from error propagation of the

components in (a), (b) and (c). $CO_2$ mixing ratios in the top panel are only shown from times

matching the radiocarbon sampling at Beromünster tower.






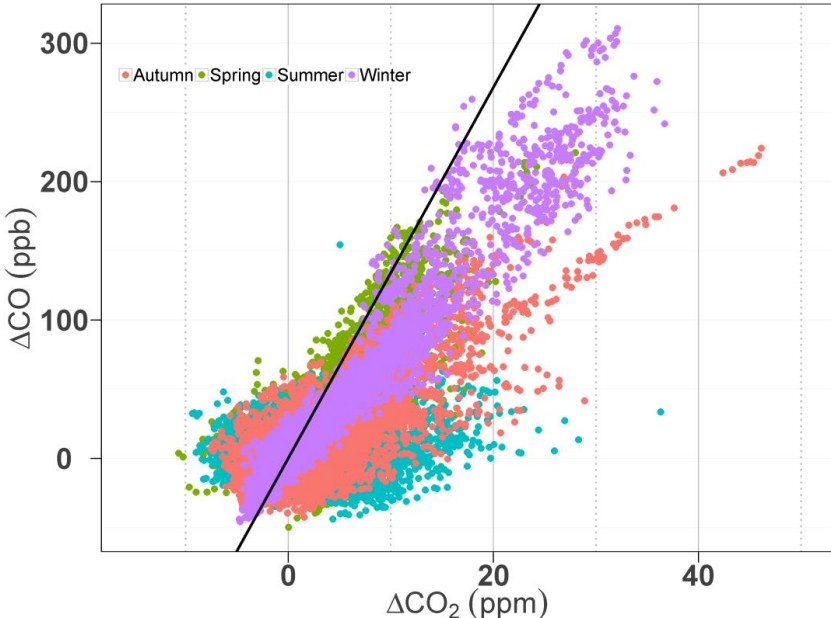


Figure 3. The correlation between enhancements in CO and $CO_2$ at Beromünster over

Jungfraujoch background for the different seasons. The black solid line shows the wintertime

$R_{CO}$ derived from radiocarbon measurements.





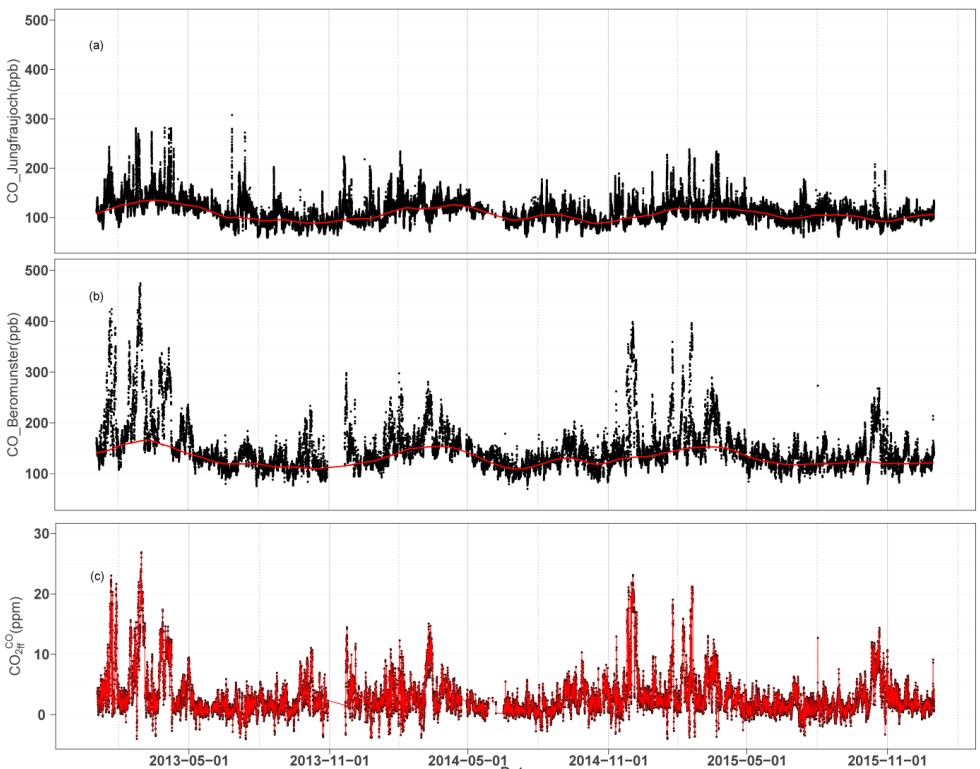


Figure 4. Time series of hourly mean CO mixing ratios measured at Jungfraujoch (a) and
Beromünster (b) sites with the red curve showing the estimated background values using the
REBS method with 60 days window. Panel (c) shows the hourly mean $CO_{2ff}$ time series
calculated using the emission ratios determined from radiocarbon measurements, and the CO
enhancements at Beromünster over the Jungfraujoch background based on Eq. (6).










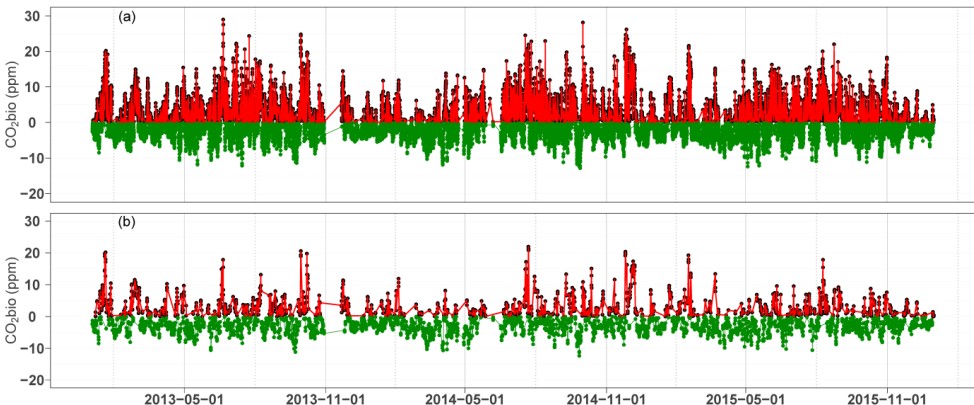


Figure 5. Time series (hourly resolution) of the biospheric $CO_2$ derived as a residual of the
difference between the total $CO_2$, $CO_{2bg}$ and $CO_{2ff}$ for all data (a), and only afternoon data
from 12:00-15:00 UTC (b). The green lines show negative $CO_{2bio}$ implying uptake while red
ones represent positive $CO_{2bio}$. The average uncertainty of $CO_{2bio}$ amounts ±1.3 ppm
calculated from error propagation.




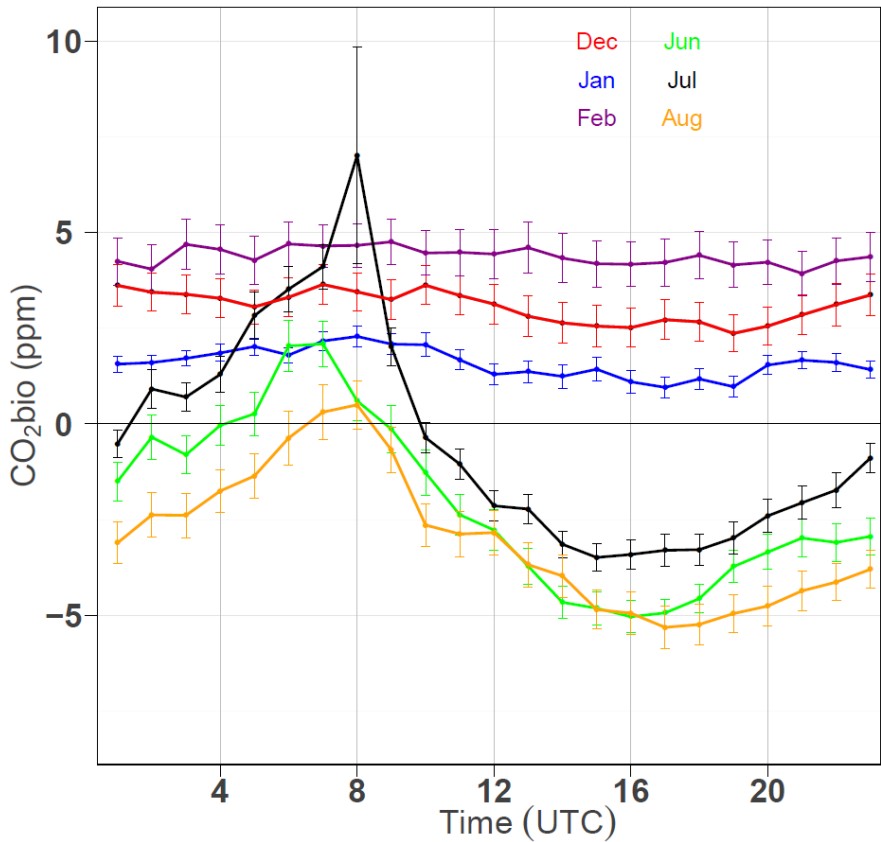


Figure 6. Hourly variations of monthly averaged biospheric $CO_2$ during summer (Jun – Aug)
and winter (Dec – Feb). While winter values dominated by respiration are constant throughout
a day, summer values show a significant diurnal variation induced by photosynthesis and
vertical mixing. The error bars are the standard deviations of the hourly averaged $CO_{2bio}$
values for each month.







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
