# Peer review of "Estimation of the fossil-fuel component in atmospheric CO2 based on radiocarbon measurements at the Beromünster tall tower, Switzerland"

_Atmospheric Chemistry and Physics, 2017_

## Referee Comment (RC1) · J. Turnbull (Referee) · 17 Apr 2017

**Jocelyn Turnbull April 14, 2017**

This paper describes a two-year times series of CO2, CO and 14CO2 measurements from a rural tall tower in Switzerland, and examines enhancements in these species relative to a Jungfraujoch high altitude (nominally free troposphere) background, including calculating fossil fuel CO2 using the 14CO2 observations. They clearly demonstrate that both fossil fuel and biogenic signals contribute to the observed CO2 enhancement over their chosen background. They observe higher CO to CO2ff ratios than might have

been expected from Swiss emissions, and also see a significant difference between the CO to CO2ff and the CO to CO2 ratios.

This paper is well written, clear and easy to follow, and well organized into sensible sections. The methodology and data quality appear to be sound (although note a few specific comments). The results are interesting and well worthy of publication in ACP, however, some additional effort is needed to sufficiently interpret many of the results, particularly the CO emission ratios. A major point is that the authors implicitly assume that they are observing Swiss sources, whereas the choice of high altitude background means that they are likely observing a much larger European footprint, which changes much of the interpretation. For this reason, I recommend major revision, but expect that the revisions should be straight-forward.

Specific comments: Abstract line 18. Please specify here that 212.5 m is the highest sampling level on the tower.

Line 69. 14C production is not only in the lower stratosphere, perhaps "upper atmosphere" would be a better phrase.

Line 72. Please reference the papers that observed and explain this trend.

Lines 82 to 84. Clarify why the biosphere is enriched in 14C relative to the atmosphere (bomb 14C that was absorbed is now being released back to the atmosphere). Also, should be "bomb tests" and "nuclear industries", not "the bomb tests" and "the nuclear industries".

Line 85. "separation" not "to separate".

Line 86. "better constrains" not "to better constrain"

Line 89. I think you mean the uncertainty in the contribution of other sources such as nuclear power.

Line 95. Turnbull et al. 2014 doesn't talk about additional tracers. Perhaps you meant
to refer to Turnbull et al 2006 or 2011?

Lines 98 to 104. Variability in the source emission ratio of CO to CO2 (or CO2ff) is likely a very important contribution. This may cause both spatial and temporal variations, e.g. Turnbull et al., 2015, Vogel et al., 2010.

Lines 165 to 167. Is this leakage at the point of sample collection (at the tower), or do you mean leakage that occurred when extracting the CO2 in the lab? Please clarify. If the leak was during the CO2 extraction, then the blank tests described in lines 170-171 are sufficient, but if the leak occurred during air collection at the tower, how can you be sure that the problem was completely resolved, and that there isn't a small remaining leak (small enough that it wouldn't be obvious from your CO2 mixing ratio calculation, but perhaps still large enough to influence the 14C results)?

Section 2.3. 14C measurement. Some more detail of the 14C data quality should be included in this section: How was the reported 14C uncertainty determined? Simply from 14C counting statistics, or is some measure of uncertainty from sample processing or long-term repeatability considered as well? Are any reference materials measured for guality control (other than the primary standards and blanks), to assess the short and long-term data guality of the atmospheric samples? At least part of the sample preparation could be assessed by examining the scatter of the 3 targets of each material averaged to determine the final 14C content. Is the 13C correction done using online AMS 13C values, or using offline IRMS 13C values? Previous work has shown quite clearly that using offline IRMS 13C values can cause substantial wheel-to-wheel biases in AMS 14C results (eg Graven et al 2007) and even with online correction, wheel-towheel variability may be observed. Is the offset between your measurements and the Heidelberg lab's measurements based entirely on the small (5 sample) intercomparison exercise described by Hammer et al 2016? It seems rather bold to assume that a single set of measurements (presumably in a single wheel) is sufficient to characterize the inter-laboratory offset over the long-term. This is a key point, since an offset of 2.1 % in  $\triangle 14C$  translates to a bias in calculated CO2ff of 0.8 ppm, or 20% of the mean ob-
served CO2ff signal! And which direction is the offset? Further, in the Hammer paper, the individual labs are held anonymous. It would be appropriate to identify which lab number in the Hammer paper corresponds to your facility (so that readers can see for themselves the offset and it's variability).

Line 196. I believe Meijer and Zondervan (1996) did a similar analysis earlier than the Levin et al work.

Lines 216-223. The authors should also refer to Turnbull et al (2009) and Miller et al (2012) where the heterotrophic respiration bias is discussed in more detail and spatially and temporally explicit bias corrections are estimated. It may be clearer to write equations 3 and 4 in a different form to clarify how the corrections (heterotrophic respiration and NPP) that are applied.

Lines 231-233. Be clear that the very large contribution identified by Vogel et al is for a reactor that is very close to the measurement site, and that reactor in question is a CANDU type, which are known to produce much more 14C than almost any other reactor type.

Lines 237-250. When measuring compared to the Jungfraujoch background, which is essentially a free troposphere background, your observations will represent some sort of continental-scale signal, not just the local signal. Is the model domain used to determine the NPP correction sufficient to capture all the NPP emissions that might be observed in your observations? Please include some discussion of this. A figure showing the model domain and influence functions would be helpful.

Lines 251-256. Are you able to provide the NPP 14C emission data (perhaps in the supplementary material)? I can imagine this would be of interest to some readers.

Equation 5. Please explain more clearly how this value is calculated. How is dCff determined in this case? And what is Anuc?

Section 2.4.3. CO:CO2 ratios. The choice of Jungfraujoch as background could be
problematic for CO, since CO has a relatively short lifetime of 1-2 months, the free troposphere may be depleted in CO relative to surface sites, biasing the ratio high.

Lines 281-283. Production of CO from oxidation of VOCs may be important too.

Lines 299-302. You say here that spikes in CO2 happened mainly in winter, but what stands out in the figure is the large and highly variable CO2 values in the summer of 2015!

Lines 303 – 305. Please show the raw 14C data as well as the corrected data. For CO and CO2, you make smooth curve fits to the Jungfraujoch background data, but it appears that the Jungfraujoch 14C data is used without smoothing. From figure 4, it is clear that Jungfraujoch experiences significant periods of polluted air – how would the results change if a smooth curve fit was used for the Jungfraujoch 14C background, or if another site was used (e.g. Niwot Ridge, which is essentially similar to Jungfraujoch in 14C, but the nature of the 14C sampling there allows exclusion of pollution events to reveal a clean air signal)?

Lines 319-320. How do you account for uncertainty in the interlaboratory offset, biosphere and NPP corrections? These are large corrections, so some measure of the uncertainty in these values should be propagated into the final CO2ff uncertainty.

Lines 322-332. This is a nice demonstration of the influence of farfield emissions – can you show the modelled influence function or trajectory?

Lines 340-343. These wildly varying CO2bio values during summer 2015 are quite peculiar and need further investigation and explanation. Is there any chance of a CO2 instrument problem during this time? It is hard to imagine how a swing from +10 to -10 ppm and back again in CO2bio could occur over a short period simply due to harvesting, and some more thought should be put into possible explanations for this.

Section 3.2. I was concerned that the use of Jungfraujoch background for CO would bias RCO high, but your comparison with using a constructed background from
Beromünster gives me some confidence that this is not a significant problem, although it could be the explanation for the slight difference in RCO for the two difference background choices.

The comparison with bottom-up inventories in section 3.3 needs to be done using RCO, not the CO:CO2 ratio, unless you also include a biogenic CO2 bottom-up estimate.

To interpret the observed RCO, you need to also consider: 1. How large are the non-fossil sources of CO for Switzerland and Europe (eg wood burning, VOC oxidation)? Could these explain the higher-than-expected RCO? Wood burning tends to be inefficient and produce high RCO values. 2. The effective footprint that the tower is "seeing" is crucial to interpreting the observed enhancements and ratios. The choice of Jungfraujoch as background means that the effective footprint is likely to include much of Europe, and certainly a much larger area than just Switzerland. Some estimate of the actual footprint should be made, either for each individual sample (or a few example days), or at least a generalized footprint, and consider the emission sources for the whole footprint. I'd suspect that RCO would be much higher in other parts of Europe and could be causing the higher-than-expected ratios. 3. You may also want to compare with Popa et al (2014) for recent Swiss traffic RCO.

Section 3.3 This section needs significant revision to address the key point that CO:CO2 ratios (as opposed to CO:CO2ff) ratios incorporate all CO2 sources and are therefore not directly related to only anthropogenic sources. In summer, and possibly spring and autumn, this ratio is essentially nonsensical, since the biogenic fluxes strongly dominate over the fossil fuel CO2 flux. Miller et al (2012) showed very clearly that even though CO:CO2 correlations may be strong in winter, they give very different slopes than CO:CO2ff correlations. Several studies have demonstrated that at the continental scale, the biogenic and fossil fuel fluxes are roughly equal in magnitude (Miller et al 2012, Turnbull et al 2015, Turnbull et al 2011b). This can readily explain the observed difference between CO:CO2 and CO2ff, and indeed the calculated CO2ff and CO2bio values also show this. In this light, the comparison of bottom-up anthropogenic

**ACPD**
CO:CO2 emission ratio should be with the observed CO:CO2ff ratio; bottom-up biogenic CO2 flux estimates are also needed to makes the comparison with observed CO:CO2 valid.

Section 3.4. Some assessment of the uncertainty in applying a single RCO value should be included here. It seems quite likely that variability in RCO through time could be driving the apparently odd results shown in figure 5. It not simply diurnally varying RCO values (as mentioned in line 418), but likely variability depending on the varying atmospheric transport bringing different emission sources (with possibly wildly varying RCO) to the site.

I don't understand how you get to figure 6 from figure 5. Figure 5 looks like a jumble of random noise, yet figure 6 looks clean and interpretable. Is figure 6 just averaging of the data shown in figure 5? The text states that during winter, the CO2bio values are mostly close to zero or positive; this is not at all obvious from figure 5. It may simply be that the scale of figure 5 makes it difficult to see what is going on. Perhaps taking a small (1 week?) section and zooming in on it would make it clear.

At the end of this section, there's a discussion of differences between different levels of the tower, but no data is shown or discussed from the lower levels. Be clear as to what data is used here and if it is CO2 data from different levels of the tower, that data should be discussed in the methods section.

Figure 2. Please show the raw 14C data as well as the corrected data. And if possible, include the actual measured values as supplemental material or point to the data archive.

Figure 3. Show the CO:CO2ff values as points, not just the line of fit.

Figure 4. Please add the 14C-based CO2ff values as points on plot c.

Figure 5. Please add a plot to show a short time period (a couple of example weeks) so that the reader can clearly see the diurnal variability. This plot is hard to look at
because there are so many data points scrunched together.

---

## Referee Comment (RC2) · Anonymous Referee #2 · 12 Jun 2017

General comments

Berhanu et al. report co-located observations of continuous CO2, CO and 14CO2 from grab samples at the Beromunster tower (Switzerland) for 2013-2016. The variability of the mixing ratio gradients relative to the high-alpine research station Jungfraujoch is discussed, especially focussing on the CO2 offset, which is demonstrably affected by both biogenic and fossil fuel CO2 fluxes. Seasonal, episodic and diurnal variations of DCO2,ff and DCO2, bio are interpreted as well as RCO, which is compared to reported emission ratios for Switzerland. The paper is well written, soundly structured and the

experimental methods are well described. The tools used to interpret the data are commonly used in this field and the results are presented clearly. Unfortunately, the interpretation of the reported results falls short at several occasions. Although plausible, more care has to be given to substantiate the interpretations.

1.) At several occasions local temperatures are given as likely cause of e.g. large positive DCO2, bio due to high temperatures (L341) or high RCO due to unusually cold conditions (L371). If T is such a strong predictor it should be added to the figures or a regression analysis added to the manuscript. The potential impact of PBL variations on reported mixing ratio gradients is also mentioned, but no thorough analysis is performed.

2.) Changes in the area of influence (footprint) are also mentioned as likely causes for specific excursions of the time series, but footprints are not given in the manuscript (or as a supplement).

3.) More broadly, the influence of a (changing) footprint seems to be ignored when the observational RCO is compared to RCO reported in the Swiss emission inventory. Before suggesting that the Swiss emission inventory potentially under-reports CO/CO2 emission ratios, the author need to demonstrate if the observed RCO is representative of average Swiss emissions or how much (and when) RCO is indeed affected by CO ad CO2 emissions from other regions (as mentioned in e.g. L371) plus photochemistry during the trajectory. Overall, the topic is of interest and novel data that could help constrain biogenic CO2 fluxes for (central) Europe can be expected to be of some interest for the ACP readership, if the major comments are addressed.

Specific comments:

L69: The 14C produced in the lower stratosphere is definitely of great interest as it is most easily transported into the troposphere, but 14C is (also/mostly) produced in other altitudes

L165: How did you ensure that samples taken before the leakage was detected were not (slightly) contaminated? Leak sometimes slowly increase over time before they are noticed or was there an abrupt change in mixing ratios or an identifiable mechanical failure?

L167: Please correct to "replaced"

L203: Please note in the text that equation 2 is only an approximation. For the correct mass-balance for 14CO2 small deltas need to be used (big delta includes an isotopic correction term based on small delta 13C)

L222: Here you mention that the correction for 14CO2, bio used cannot (fully) account for short-term respiration changes, yet the daily cycle of CO2, bio is discussed (see Figure 6). Please include a comment to which degree the choice of a simple correction could alter the retrieved CO2,bio in the results/discussions section

L256: Please expand why the 2015 14C emissions Benzau emissions were assumed to be 0 during the shut-down period. The production of radionuclides should be smaller during maintenance, but more possibilities of contamination or release might exist (depending on reactor type and maintenance/intervention)

L274: Why is precision of 10-min aggregates reported for JFJ, while long-term reproducibility was reported for Beromunster observations? How are those two quantities combined into one uncertainty for DCO?

L329: Please consider highlighting periods with southeastern European air masses in Figure 2.

L349: Please clarify: if the reported uncertainties of RCO (summer and winter) in L347 and L348 are correct there is no significant seasonal difference. Hence, the authors should discuss why there is no difference rather than discuss a reason for a non-existing seasonality.

L355: Consider changing to: "The value obtained this way is statistically not different

. . ." This could also be discussed more in terms of its implications.

L371: The authors mention that cold conditions and mass transport from Eastern Europe are likely causes, yet no meteorological data is shown in this paper. Please consider adding a supplement with the key information that you have based this analysis on.

L380: The ratios do indeed differ significantly, but you need to establish why Beromunster-JFJ based RCO should be representative for Switzerland (only). See general comment #3

L407: Colder temperatures are mentioned a main cause, again. If temperature is such a good predictor of DCO2ff a simple scatter plot should suffice to strengthen your argument. Likely an analysis of the impact of the PBL would be useful.

L425: Please consider adding visual aides to highlight the seasonal cycle in Figure 5. It seems not too clear in the printed version.

Table 1: An overall of 45 RCO values is reported, while the study is supposedly based on a 3-years long time series (L451) of bi-weekly samples (i.e. 78). Seven samples were accounted for due to the leakage problem reported in L167. What caused the other 26 to be missing here? Were they excluded, not samples, etc?

Figure 2: Do the dashed lines in 2c and 2d both denote the averages of is the dashed line in 2d just y=0ppm

Figure 5: see comment L425

---

## Author Comment (AC1)

We would like to thank J. Turnbull for her helpful and supportive comments and below we have addressed the comments/questions. For clarity, we keep the editors comments in blue and our replies are in black font.

This paper describes a two-year times series of CO2, CO and 14CO2 measurements from a rural tall tower in Switzerland, and examines enhancements in these species relative to a Jungfraujoch high altitude (nominally free troposphere) background, including calculating fossil fuel CO2 using the 14CO2 observations. They clearly demonstrate that both fossil fuel and biogenic signals contribute to the observed CO2 enhancement over their chosen background. They observe higher CO to CO2ff ratios than might have been expected from Swiss emissions, and also see a significant difference between the CO to CO2ff and the CO to CO2 ratios.

This paper is well written, clear and easy to follow, and well organized into sensible sections. The methodology and data quality appear to be sound (although note a few specific comments). The results are interesting and well worthy of publication in ACP, however, some additional effort is needed to sufficiently interpret many of the results, particularly the CO emission ratios. A major point is that the authors implicitly assume that they are observing Swiss sources, whereas the choice of high altitude background means that they are likely observing a much larger European footprint, which changes much of the interpretation. For this reason, I recommend major revision, but expect that the revisions should be straight-forward.

Specific comments: Abstract

line 18. Please specify here that 212.5 m is the highest sampling level on the tower.

We have now added the phrase "… from the highest sampling inlet (212.5 m)… "

Line 69. 14C production is not only in the lower stratosphere, perhaps "upper atmosphere" would be a better phrase.

We agree that "upper atmosphere" fits better and replaced accordingly.

Line 72. Please reference the papers that observed and explain this trend.

We have now added the following references: (Manning et al., 1990; Levin et al., 2010)

Lines 82 to 84. Clarify why the biosphere is enriched in 14C relative to the atmosphere (bomb 14C that was absorbed is now being released back to the atmosphere).

We have rephrased the sentence on lines 83-87 as follows:

However, this depletion can also partially be offset by $CO_2$ release from the biosphere which has enriched $^{14}C/^{12}C$ ratios due to nuclear bomb tests in the 1960's. $^{14}C$ produced by these tests was absorbed by the land biosphere and it is now gradually being released back to the atmosphere (Naegler and Levin, 2009). Another contribution could be direct 14C emissions from nuclear industries (Levin et al., 2010).

Also, should be "bomb tests" and "nuclear industries", not "the bomb tests" and "the nuclear industries".

Changed as suggested.

Line 85. "separation" not "to separate".

Replaced accordingly.

Line 86. "better constrains" not "to better constrain"

Replaced accordingly

Line 89. I think you mean the uncertainty in the contribution of other sources such as nuclear power.

Yes, we have now rephrased this sentence as "…as well as the uncertainty in the contribution from other sources of $^{14}C$ such as nuclear power plants (NPPs)"

Line 95. Turnbull et al. 2014 doesn't talk about additional tracers. Perhaps you meant to refer to Turnbull et al 2006 or 2011?

Thank you for pointing out. We have replaced Tunrbull et al. 2014 with Turnbull et al. 2006 and Turnbull et al. 2011

Lines 98 to 104. Variability in the source emission ratio of CO to CO2 (or CO2ff) is likely a very important contribution. This may cause both spatial and temporal variations, e.g. Turnbull et al., 2015, Vogel et al., 2010.

We have now added this information in lines 106-108:

Additionally, variability in the $CO/CO_2$ emission ratios of the sources can contribute to its spatial and temporal variability (Vogel et al., 2010; Turnbull et al., 2015; Oney et al., 2017).

Lines 165 to 167. Is this leakage at the point of sample collection (at the tower), or do you mean leakage that occurred when extracting the CO2 in the lab? Please clarify. If the leak was during the CO2 extraction, then the blank tests described in lines 170-171 are sufficient, but if the leak occurred during air collection at the tower, how can you be sure that the problem was completely resolved, and that there isn't a small remaining leak (small enough that it wouldn't be obvious from your CO2 mixing ratio calculation, but perhaps still large enough to influence the 14C results)?

The blank test was conducted to check the leakage at the extraction line. However, the leakage occurred at the tower site during sample collection. After replacement of all the exhaust pumps, we started conducting regular leakage tests by closing the needle valves before the pumps and checking in case any flow (leakage) is present with the flow meter in front of the pump but we have not observed any rise in pressure ensuring no more leakage.

We have added the above sentences in lines 170-181.

Section 2.3. 14C measurement.

Some more detail of the 14C data quality should be included in this section: How was the reported 14C uncertainty determined? Simply from 14C counting statistics, or is some measure of uncertainty from sample processing or long-term repeatability considered as well? Are any reference materials measured for quality control (other than the primary standards

and blanks), to assess the short and long-term data quality of the atmospheric samples? At least part of the sample prepa- ration could be assessed by examining the scatter of the 3 targets of each material averaged to determine the final 14C content. Is the 13C correction done using online AMS 13C values, or using offline IRMS 13C values? Previous work has shown quite clearly that using offline IRMS 13C values can cause substantial wheel-to-wheel biases in AMS 14C results (eg Graven et al 2007) and even with online correction, wheel-to-wheel variability may be observed. Is the offset between your measurements and the Heidelberg lab's measurements based entirely on the small (5 sample) intercompari- son exercise described by Hammer et al 2016? It seems rather bold to assume that a single set of measurements (presumably in a single wheel) is sufficient to characterize the inter-laboratory offset over the long-term. This is a key point, since an offset of 2.1‰ in Δ14C translates to a bias in calculated CO2ff of 0.8 ppm, or 20% of the mean observed CO2ff signal! And which direction is the offset? Further, in the Hammer paper, the individual labs are held anonymous. It would be appropriate to identify which lab number in the Hammer paper corresponds to your facility (so that readers can see for themselves the offset and it's variability).

The green dataset in Fig. 2b represents an average of 6 individual $^{14}$C measurements, *i.e.* from the three individual samples that were collected over a 15-minute interval (see Section 2.2) in duplicate each (see Section 2.3). The uncertainty of an individual $^{14}$C measurement typically amounts ~2.1‰, including contributions from counting statistics (~1.1‰), corrections of normalization (*i.e.* blank subtraction, standard normalization, and correction for isotopic fractionations) (~1.1‰) and an unaccounted long-term variability of sampling and $^{14}$C analysis according to Szidat et al., 2014 (1.5 ‰) (Szidat et al., 2014), contributions comparable to previous observations (Graven et al., 2007).

During calculation of weighted averages of the duplicates, the uncertainty of the mean is determined with the contributions of the counting statistics and the normalization, whereas the uncertainty of the unaccounted long-term variability is considered fully afterwards, as this contribution cannot be reduced by averaging of two measurements performed on the same day. This uncertainty of the weighted average typically amounts ~1.9‰; it is compared with the standard deviation of the duplicates and the larger of these values is used as the final uncertainty of the duplicates. The mean of the three individual samples from the same day, which is used in Section 2.4.1 as $\Delta^{14}C_{meas}$, is then determined and associated with the average uncertainty of the three duplicates, as the variability of the three samples is comparable to this average uncertainty for all cases.

For the fractionation correction, $\delta^{13}$C values of the AMS are used, which show a long-term standard uncertainty of ±1.2‰ (Szidat et al., 2014). The AMS $\delta^{13}$C values agree well on the average with corresponding IRMS results, revealing a statistically insignificant difference of -0.2±1.2‰ with slightly more depleted AMS results. A wheel-to-wheel dependence of this difference is not observed.

Indeed, the interlaboratory correction between the AMS laboratory at Bern and the low-level counting (LLC) laboratory at Heidelberg is based on the results of the data given by Hammer et al., 2017. The LARA lab code in this intercomparison is #2, whereas Heidelberg is indicated as LLC. The measurement bias (*i.e.* the mean difference of the measured $\Delta^{14}$C minus the consensus value of the participating laboratories for all investigated $CO_2$ samples) is +1.8±0.1‰ and -0.3±0.5‰ for Bern and Heidelberg, respectively, from which the bias

between both labs of 2.1±0.5‰ is determined with a larger measured $\Delta^{14}C$ for Bern. We acknowledge that the number of five samples used in this intercomparison is too small to quantify satisfactorily the bias between both radiocarbon labs. Hammer et al. (2017) estimated that approximately 50 samples would be needed to fulfill this goal and state that a second stage of this intercomparison is planned with this number of samples, but a reduced number of AMS labs. The Bern LARA lab has registered its large interest to participate in this second stage. Until then, however, the correction of the interlaboratory bias relies on the existing intercomparison data.

The text in Section 2.3 has been modified accordingly.

Line 196. I believe Meijer and Zondervan (1996) did a similar analysis earlier than the Levin et al work.

We have included (Zondervan and Meijer, 1996)

Lines 216-223. The authors should also refer to Turnbull et al (2009) and Miller et al (2012) where the heterotrophic respiration bias is discussed in more detail and spatially and temporally explicit bias corrections are estimated. It may be clearer to write equations 3 and 4 in a different form to clarify how the corrections (heterotrophic respiration and NPP) that are applied.

We added Eq. 5 to include the corrections for heterotrophic respiration and NPPs following Eq. 6 in Turnbull et al. (2009) and the following section is added to these lines (254-262):

However, the $CO_{2ff}$ determined using Eq. (4) incorporates a small bias due to the non-negligible disequilibrium contribution of heterotrophic respiration as well as due to contributions from NPPs. To correct for the bias from these other contributions, an additional term ($CO_{2other}$ and $\Delta^{14}C_{other}$) can be included in Eq. (4) as suggested by Turnbull et al. (2009) and Miller et al. (2012):

$$CO_{2ff} = \frac{CO_{2meas}(\Delta^{14}C_{bg} - \Delta^{14}C_{meas})}{\Delta^{14}C_{bg} + 1000‰} + \frac{CO_{2other}(\Delta^{14}C_{other} - \Delta^{14}C_{bg})}{\Delta^{14}C_{bg} + 1000‰} \quad (5)$$

Lines 231-233. Be clear that the very large contribution identified by Vogel et al is for a reactor that is very close to the measurement site, and that reactor in question is a CANDU type, which are known to produce much more 14C than almost any other reactor type.

Thank you for pointing this out. We extended this sentence on lines 281-283 as:

"…though this large number was obtained for a site in close vicinity of a CANada Deuterium Uranium (CANDU) type reactor known for producing particularly high $^{14}C$ emissions."

Lines 237-250. When measuring compared to the Jungfraujoch background, which is essentially a free troposphere background, your observations will represent some sort of continental-scale signal, not just the local signal. Is the model domain used to determine the NPP correction sufficient to capture all the NPP emissions that might be observed in your observations? Please include some discussion of this. A figure showing the model domain and influence functions would be helpful.

This is correct. Our model domain (as already stated in Section 2.4.2) covers large parts of Western Europe and thus we do not only simulate the contribution from Swiss NPPs but also

those in other countries, especially in France. Note that on line 285 we already mentioned the potential influence of French NPPs, and on line 380 we mentioned that about 70 % of the enhancements were due to Swiss NPPs and 30 % from foreign NPPs.

To make this clearer, we changed the beginning of the sentence on line 287 from "To estimate the influence of NPPs…" to "To estimate the influence of Swiss and other European NPPs…" and we changed the description of the model domain in line 292 to "for a domain covering large parts of Western Europe from the southern tip of Spain to the northern tip of Denmark and from the western coast of Ireland to eastern Poland."

Lines 251-256. Are you able to provide the NPP 14C emission data (perhaps in the supplementary material)? I can imagine this would be of interest to some readers.

Annual totals of 14C emissions are published in the annual reports of the Swiss Federal Nuclear Safety Inspectorate ENSI (https://www.ensi.ch/de/dokumente/document-category/strahlenschutzberichte/). This reference is added. However, we are not allowed to provide the full time series that we used in the publication.

Equation 5. Please explain more clearly how this value is calculated. How is dCff determined in this case? And what is Anuc?

Indeed, we missed to explain $\delta A_{nuc}$, which is the enhancement in $^{14}CO_2$ mole fractions due to NPPs at Beromünster ($A_{nuc}$) relative to $^{14}CO_2$ mole fractions due to NPPs at the reference site JFJ ($A^R_{nuc}$). Similarly, $\delta C_{ff}$ is the enhancement in fossil fuel $CO_2$ at Beromünster relative to the fossil fuel $CO_2$ at JFJ. Actually, lacking transport simulations for Jungfraujoch and hence the reference values we ultimately chose to start from the simpler Eq. (4) of Levin et al. (2010):

$$\Delta^{14}C = f \ n^{14}/n^C - 1000$$

with the dimensionless factor $f = 8.19 \times 10^{14}$ and $n^{14}/n^C$ being the number of $^{14}C$ atoms relative to the total number of C-atoms ($^{12}C + ^{13}C + ^{14}C$). The equation is obtained when a constant $\delta^{13}C$ value of $-7$ ‰ is assumed (Levin et al., 2010). The $\Delta^{14}C$ contribution due to NPPs at Beromünster was then estimated by taking the ratio of the mole fraction of $^{14}C$ due to NPPs ($n^{14}_{npp}$) simulated with FLEXPART-COSMO to the mole fraction of $CO_2$ ($n^{CO2}_{meas}$) measured at Beromünster in the above equation. If we further assume that the $n^{14}_{npp}$ signal at the reference site (Jungfraujoch) is zero (or very small relative to the one at Beromünster), then we obtain the difference $\delta\Delta^{14}C$ in $\Delta^{14}C$ due to NPPs between the two sites as:

$$\delta\Delta^{14}C = f \, n^{14}_{npp}/n^C_{meas}$$

which is included in the revised manuscript.

Applying this equation instead of Graven and Gruber (2011) results in a 3.7 % lower value, since $f$ is 3.7 % lower than the factor $1000/R_s$ of their study.

Section 2.4.3. CO:CO2 ratios

The choice of Jungfraujoch as background could be problematic for CO, since CO has a relatively short lifetime of 1-2 months, the free troposphere may be depleted in CO relative to surface sites, biasing the ratio high.

This is a valid point and we agree with the reviewer that the choice of background could

affect the result. However, shown in section 3.2, the choice of background had only a minimal effect on the derived emission ratios.

Lines 281-283. Production of CO from oxidation of VOCs may be important too.

It is true that production of CO from the oxidation of VOCs could be important, an issue that was further elaborated in Oney et al. (2017). However, since we are focusing in this paragraph on comparing the ratios $R_{CO}$ and $CO/CO_2$ and since the ratios will be affected in a similar way, we chose not mention this issue here.

Lines 299-302. You say here that spikes in CO2 happened mainly in winter, but what stands out in the figure is the large and highly variable CO2 values in the summer of 2015!

Please see the comments below on page 8.

Lines 303 – 305. Please show the raw 14C data as well as the corrected data.

We have now included the data before correction for contribution from NPPs and heterotrophic respiration disequilibrium as green data points in Figure 2b also shown below.

[Figure]

Figure 2b. $\Delta^{14}C$ determined from the bi-weekly point samplings at the site before (green) and after (red) correction for the intercomparison offset (see section 2.3) and the $^{14}C$ contribution from NPPs (see Eq. 5) and from 14-days integrated samplings at Jungfraujoch (blue).

For CO and CO2, you make smooth curve fits to the Jungfraujoch background data, but it appears that the Jungfraujoch 14C data is used without smoothing. From figure 4, it is clear that Jungfraujoch experiences significant periods of polluted air – how would the results change if a smooth curve fit was used for the Jungfraujoch 14C background, or if another site was used (e.g. Niwot Ridge, which is essentially similar to Jungfraujoch in 14C, but the nature of the 14C sampling there allows exclusion of pollution events to reveal a clean air signal)?

If we derive a smoothed curve for the $^{14}C$ measurements at JFJ and compare the calculated $CO_{2ff}$, the difference is minor. We have calculated a slope of 1.0003 ($r^2 = 0.944$) for a linear correlation between $CO_{2ff}$ determined with and without smoothed data.

In case of $R_{CO}$, smoothing did not change the slope but the correlation coefficient showed a slight increase to 0.7 from the previous value of 0.6. This information is added on lines 405-406 in the revised manuscript.

Lines 319-320. How do you account for uncertainty in the interlaboratory offset, biosphere and NPP corrections? These are large corrections, so some measure of the uncertainty in these values should be propagated into the final CO2ff uncertainty.

The uncertainty in $CO_{2ff}$ determined from $^{14}C$ measurements is composed of uncertainties in $^{14}C$ measurements, choice of background and additional bias from other sources such as NPPs (Turnbull et al., 2009).

The measurement uncertainty in $CO_{2ff}$ is composed of the measurement uncertainty in $^{14}C$ for both the measurement and the background site (2.0 ‰).

An uncertainty of 0.3 ppm in the bias from biospheric $^{14}C$ contribution is included based on the work of Turnbull et al. (2009).

The uncertainty of the interlaboratory offset has been documented in lines 219-229: A recent interlaboratory compatibility test estimated a small bias of 2.1 ± 0.5 ‰ (Hammer et al., 2016) between the two institutes, which was subsequently subtracted from the $^{14}C$ measurements of the Beromünster samples.

Consequently, in Equation (5), the error propagation of $\Delta^{14}C_{meas}$ included a term from the analysis (typically ~ 2.0 ‰) and a term for the interlaboratory offset (0.5 ‰) so that overall uncertainty of $\Delta^{14}C_{meas}$ is increased by <0.1‰ due to the interlaboratory offset.

The uncertainty of the contribution from NPPs is very difficult to judge as it includes both uncertainties in the transport simulation and temporal fluctuations in $^{14}C$ emissions from NPPs not resolved by the bi-weekly to monthly measurements. Angevine et al. (2014) have estimated an uncertainty of Lagrangian transport simulations of 30-40% using an ensemble of different meteorological forcings. This uncertainty seems reasonable also in our case considering the good agreement between measured and a posteriori simulated methane concentrations at Beromünster reported by Henne et al. (2016). The uncertainty due to unaccounted temporal variability in the source is roughly estimated to 50%, based on the standard deviation of the bi-weekly measurements at Muehleberg which is 34% of the annual mean. Addition of the two squared uncertainties leads to an overall estimated uncertainty of about 60-70% (~1 to 1.2 ‰ for a mean $^{14}C$ contribution of 1.77 ‰ at this site)

Lines 322-332. This is a nice demonstration of the influence of farfield emissions – can you show the modelled influence function or trajectory?

All individual footprints can now be looked up in the supplement. The one for 27 March 2014 is shown below, which demosntrates a strong sensitivity to the area of Zurich (largest population center in Switzerland) as well as advection from eastern Europe.

[Figure]

Figure 1. FLEXPART-COSMO simulated foot print corresponding to a radiocarbon sampling on 27 March 2014.

Lines 340-343. These wildly varying CO2bio values during summer 2015 are quite peculiar and need further investigation and explanation. Is there any chance of a CO2 instrument problem during this time? It is hard to imagine how a swing from +10 to -10 ppm and back again in CO2bio could occur over a short period simply due to harvesting, and some more thought should be put into possible explanations for this.

During summer 2015, we observed strong variability in both $CO_2$ and $CO_{2bio}$ and we had no problem with the measurement system. However, this period was one of the hottest and driest summers in central Europe (Orth et al., 2016). In Switzerland, it was the second hottest summer since the beginning of measurements in 1864 with most of the extreme dates in July (MeteoSuisse, 2015). Such climate extremes can lead to enhanced respiration and reduced photosynthesis, in turn, higher $CO_2$ and $CO_{2bio}$ in the atmosphere. Looking specifically at the two data points in June and July 2015, the daily average temperatures recorded at Beromünster were 24.6 °C and 26 °C at the highest inlet of 212.5 m. Based on measurements at Beromünster and other cities of the CarboCount CH network in 2013, Oney et al. (2017) reported that for a daily mean temperature of greater than 20 °C, the biosphere over the Swiss plateau tends to become a net $CO_2$ source. The observed positive spikes in $CO_2$ (Fig. 2a) and $CO_{2bio}$ (Fig. 2d) likely resulted from such extremes. These sentences are now added in lines 390 – 401.

Section 3.2

I was concerned that the use of Jungfraujoch background for CO would bias RCO high, but your comparison with using a constructed background from Beromünster gives me some confidence that this is not a significant problem, although it could be the explanation for the slight difference in RCO for the two difference background choices.

Indeed, the slight change in $R_{CO}$ from 13.4 ppb/ppm to 12.8 ppb/ppm may be due to the differences in the CO background.

The comparison with bottom-up inventories in section 3.3 needs to be done using RCO, not the CO:CO2 ratio, unless you also include a biogenic CO2 bottom-up estimate.

To interpret the observed RCO, you need to also consider: 1. How large are the non-fossil sources of CO for Switzerland and Europe (eg wood burning, VOC oxidation)? Could these explain the higher-than-expected RCO? Wood burning tends to be inefficient and produce high RCO values. 2. The effective footprint that the tower is "seeing" is crucial to interpreting the observed enhancements and ratios. The choice of Jungfraujoch as background means that the effective footprint is likely to include much of Europe, and certainly a much larger area than just Switzerland. Some estimate of the actual footprint should be made, either for each individual sample (or a few example days), or at least a generalized footprint, and consider the emission sources for the whole footprint. I'd suspect that RCO would be much higher in other parts of Europe and could be causing the higher-than-expected ratios. 3. You may also want to compare with Popa et al (2014) for recent Swiss traffic RCO.

In light of the comments above, we have modified section 3.3. The comparison of $R_{CO}$ is done only with the bottom-up inventory.

With respect to the interpretation of the high $R_{CO}$, the following points were included in section 3.2.

The $R_{CO}$ value derived in this study is significantly higher than the anthropogenic CO to $CO_2$ emission ratio of 7.8 mmol/mol calculated from Switzerland's greenhouse gas inventory report for 2013 (FOEN, 2015a, b).

As suggested by the reviewer, besides fossil fuel emissions, non-fossil contributions such as oxidation of VOCs and wood burning can play a role in the observed $R_{CO}$. The national inventory attributes about 15 % of total $CO_2$ emissions in 2014 to non-fossil fuel sources such as combustion of wood, waste incineration, and biofuels (FOEN, 2015a).

The potential contribution of oxidation of VOCs was discussed in more detail in Oney et al. (2017). Unfortunately, current literature spans a very large range of estimates of this influence which does not allow drawing firm conclusions. We expect that the contribution is relatively small in Europe (with much larger anthropogenic versus biospheric emission density when compared for example to the US), especially in winter when OH radical concentrations are small.

We fully agree that differences between Beromünster and Jungfraujoch are not only sensitive to emissions in Switzerland, although the annual mean foot print of Beromünster shows a predominant influence from the Swiss Plateau (Oney et al., 2015).

Another source of discrepancy between the two emission ratio estimates can indeed be due to enhanced CO emissions transported from other European cities towards Beromünster. Oney et al. (2017) observed particularly large $CO/CO_2$ ratios at Beromünster during several pollution events in late winter and early spring 2013 which were associated with air mass transport from eastern Europe where poorly controlled combustion of biofuels and coal likely results in high ratios.

Regarding the comparison of the observed $R_{CO}$ with the tunnel measurements by Popa et al., we have added the following paragraph in lines 418-424.

"A recent study investigating the CO to $CO_2$ ratio from road traffic in Islisberg tunnel, Switzerland also observed a significant decrease in this ratio comparing to previous estimates pointing to a substantial reduction in CO emissions from road traffic with a $CO/CO_2$ ratio of $4.15 \pm 0.34$ ppb/ppm (Popa et al., 2014). This may also indicate a significant contribution from non-road traffic emissions which accounts for more than 70 % of the total $CO_2$ emissions leading to the high apparent $R_{CO}$."

Section 3.3

This section needs significant revision to address the key point that CO:CO2 ratios (as opposed to CO:CO2ff) ratios incorporate all CO2 sources and are therefore not directly related to only anthropogenic sources. In summer, and possibly spring and autumn, this ratio is essentially nonsensical, since the biogenic fluxes strongly dominate over the fossil fuel CO2 flux. Miller et al (2012) showed very clearly that even though CO:CO2 correlations may be strong in winter, they give very different slopes than CO:CO2ff correlations. Several studies have demonstrated that at the continental scale, the biogenic and fossil fuel fluxes are roughly equal in magnitude (Miller et al 2012, Turnbull et al 2015, Turnbull et al 2011b). This can readily explain the observed difference between CO:CO2 and CO2ff, and indeed the calculated CO2ff and CO2bio values also show this. In this light, the comparison of bottom-up anthropogenic CO:CO2 emission ratio should be with the observed CO:CO2ff ratio; bottom-up biogenic CO2 flux estimates are also needed to makes the comparison with observed CO:CO2 valid.

This section has been re-written with respect to biospheric contribution even during winter now referencing to previous studies by Miller et al 2012, Turnbull et al 2011b and Turnbull et al 2015.

The main points include that CO:$CO_2$ ratios observed even in winter are strongly biased by a contribution from biospheric respiration. A similar contribution has been observed previously from air samples in East Asia amounting to 20-30% biospheric contribution during this period and this contribution can be as large as the $CO_{2ff}$ contribution (Turnbull et al., 2011; Miller et al., 2012).

**Section 3.4.**

Some assessment of the uncertainty in applying a single RCO value should be included here. It seems quite likely that variability in RCO through time could be driving the apparently odd results shown in figure 5. It not simply diurnally varying RCO values (as mentioned in line 418), but likely variability depending on the varying atmospheric transport bringing different emission sources (with possibly wildly varying RCO) to the site.

I don't understand how you get to figure 6 from figure 5. Figure 5 looks like a jumble of random noise, yet figure 6 looks clean and interpretable. Is figure 6 just averaging of the data shown in figure 5?The text states that during winter, the CO2bio values are mostly close to zero or positive; this is not at all obvious from figure 5. It may simply be that the scale of figure 5 makes it difficult to see what is going on. Perhaps taking a small (1 week?) section and zooming in on it would make it clear.

Yes, Figure 6 is derived from the same data as Figure 5 by averaging over all points of a given hour of the day and a given month. Although Figure 5 looks like random noise, the comparison between Figures 5a and 5b already shows that the variability has a diurnal component as in the afternoon the negative points (green) become more prominent. The large positive excursions in Fig. 5a typically occur at night (especially in summer) when the boundary layer is low and accumulation of nighttime respiration fluxes is large.

At the end of this section, there's a discussion of differences between different levels of the tower, but no data is shown or discussed from the lower levels. Be clear as to what data is used here and if it is CO2 data from different levels of the tower, that data should be discussed in the methods section.

A detailed comparison of the $CO_2$ measurements at all 5 sampling heights was presented by Satar et al. (2016), which was referenced in the text. We agree, however, that the sentence was confusing. It was replaced by

"As reported by Satar et al. (2016), this decrease in early morning CO2 concentrations at the 212 m inlet is lagging the decrease at the lowest sampling level of 12.5 m by approximately one hour."

Figure 2. Please show the raw 14C data as well as the corrected data. And if possible, include the actual measured values as supplemental material or point to the data archive.

We have now added the raw $^{14}$C data in Figure 2b. The actual measured data is also included in the supplementary files.

Figure 3. Show the CO:CO2ff values as points, not just the line of fit.

We have followed the suggestion and added the $\Delta CO:CO_{2ff}$ values as points in Figure 3.

Figure 4. Please add the 14C-based CO2ff values as points on plot c.

We have now added $CO_{2ff}$ values derived from radiocarbon measurements to Figure 4c.

Figure 5. Please add a plot to show a short time period (a couple of example weeks) so that the reader can clearly see the diurnal variability. This plot is hard to look at because there

are so many data points scrunched together.

It is indeed very important to see the diurnal variability in Figure 5. However, we did not include a zoom-in of the $CO_{2bio}$ for a couple of days/weeks in this figure because here the idea is to see the seasonal variability and effect of photosynthesis which is dominant during afternoon hours. In addition we have shown the diurnal variability averaged over the entire period for each hour in Figure 6. However, we have now included two randomly selected periods of three consecutive days in January and July 2014 in the supplementary materials section.

References

Angevine, W. M., Brioude, J., McKeen, S., and Holloway, J. S.: Uncertainty in Lagrangian pollutant transport simulations due to meteorological uncertainty from a mesoscale WRF ensemble, Geosci Model Dev, 7, 2817-2829, 10.5194/gmd-7-2817-2014, 2014.

[revised manuscript text omitted]

Supplementary Figures

[Figure]

Figure S1. The diurnal variability of $CO_{2bio}$ for selected two periods: 01 - 03 January 2014 and 01 - 03 June 2015.

---

## Author Comment (AC2)

**Reply to Interactive comments from the Anonymous Referee #2**

We would like to thank Anonymous Referee #2 for the helpful and supportive comments. Below we have addressed the comments/questions point-wise. For clarity, we keep the editors comments in blue and our replies are in black font.

General comments

Berhanu et al. report co-located observations of continuous CO2, CO and 14CO2 from grab samples at the Beromünster tower (Switzerland) for 2013-2016. The variability of the mixing ratio gradients relative to the high-alpine research station Jungfraujoch is discussed, especially focussing on the CO2 offset, which is demonstrably affected by both biogenic and fossil fuel CO2 fluxes. Seasonal, episodic and diurnal variations of DCO2,ff and DCO2, bio are interpreted as well as RCO, which is compared to reported emission ratios for Switzerland. The paper is well written, soundly structured and the experimental methods are well described. The tools used to interpret the data are commonly used in this field and the results are presented clearly. Unfortunately, the interpretation of the reported results falls short at several occasions. Although plausible, more care has to be given to substantiate the interpretations.

1.) At several occasions local temperatures are given as likely cause of e.g. large positive DCO2, bio due to high temperatures (L341) or high RCO due to unusually cold conditions (L371). If T is such a strong predictor it should be added to the figures or a regression analysis added to the manuscript. The potential impact of PBL variations on reported mixing ratio gradients is also mentioned, but no thorough analysis is performed.

As the reviewer pointed out, we have ascribed part of the observed strong depletions in radiocarbon or enhanced biospheric fluxes to local temperature. This latter case was also elaborated in a recent study from the same site (Oney et al., 2017). We have now included the temperature record of the site from the highest inlet of the tower (212.5 m), to match with the radiocarbon measurements in Fig. 2e.

We discussed this issue further in reply to reviewer#1 (page 8) and additional information is included in the revised manuscript lines 386-401.

We agree with the reviewer that through analysis of the PBL variations is important to better understand variations in $R_{CO}$ and $CO_{2bio}$. However, this is beyond the current scope of this study but will be considered in future studies and this information is provided in the conclusion section.

2.) Changes in the area of influence (footprint) are also mentioned as likely causes for specific excursions of the time series, but footprints are not given in the manuscript (or as a supplement).

The simulated footprints from FLEXPART-COSMO are now provided in the supplementary

section.

3.) More broadly, the influence of a (changing) footprint seems to be ignored when the observational RCO is compared to RCO reported in the Swiss emission inventory. Before suggesting that the Swiss emission inventory potentially under-reports CO/CO2 emission ratios, the author need to demonstrate if the observed RCO is representative of average Swiss emissions or how much (and when) RCO is indeed affected by CO ad CO2 emissions from other regions (as mentioned in e.g. L371) plus photochemistry during the trajectory. Overall, the topic is of interest and novel data that could help constrain biogenic CO2 fluxes for (central) Europe can be expected to be of some interest for the ACP readership, if the major comments are addressed.

The Beromünster tower, which is situated on the southern border of the Swiss plateau has been assessed for its foot print in previous publications (Oney et al., 2015; Henne et al., 2016). Both studies show that the measurements are mostly representative of the Swiss plateau (the flat part of Switzerland between the Alps and the Jura mountains where the vast majority of the population lives and where most of the industrial and agricultural activity takes place). The study by Henne et al. (2016) on inverse modelling of $CH_4$ further revealed an excellent agreement between the top-down estimates and the Swiss methane emissions further suggesting that measurements at the tower capture concentration fluctuations that are representative for emissions in Switzerland.

Specific comments:

L69: The 14C produced in the lower stratosphere is definitely of great interest as it is most easily transported into the troposphere, but 14C is (also/mostly) produced in other altitudes.

Following also the suggestion of the other reviewer, we have changed the term "lower stratosphere" to "upper atmosphere"

L165: How did you ensure that samples taken before the leakage was detected were not (slightly) contaminated? Leak sometimes slowly increase over time before they are noticed or was there an abrupt change in mixing ratios or an identifiable mechanical failure?

Indeed possible minor leaks before detection could not be ignored. However, such minor leaks are expected to manifest themselves in deviations not only in the amount of $CO_2$ that has been extracted but also in its stable isotope ratio measurements. Yet, not observed such deviations in these periods.

L167: Please correct to "replaced"

We have now corrected the word "replaced"

L203: Please note in the text that equation 2 is only an approximation. For the correct mass-balance for 14CO2 small deltas need to be used (big delta includes an isotopic correction term based on small delta 13C)

We have now modified the paragraph about equation 2 as:

Each of these components has a specific $\Delta^{14}C$ value (i.e. the deviation in per mil of the $^{14}C/^{12}C$ ratio from its primary standard, and corrected for fractionation and decay using $^{13}C$ measurements) described as $\Delta^{14}C_{meas}$, $\Delta^{14}C_{bg}$, $\Delta^{14}C_{bio}$ and $\Delta^{14}C_{ff}$. In analogy to Eq. (1), a mass balance approximation equation can also be formulated for $^{14}C$ as…

L222: Here you mention that the correction for 14CO2, bio used cannot (fully) account for short-term respiration changes, yet the daily cycle of CO2, bio is discussed (see Figure 6). Please include a comment to which degree the choice of a simple correction could alter the retrieved CO2,bio in the results/discussions section

In case we do not apply this simple correction the $CO_{2bio}$ will be changed by about 0.4 ppm on average.

L256: Please expand why the 2015 14C emissions Benzau emissions were assumed to be 0 during the shut-down period. The production of radionuclides should be smaller during maintenance, but more possibilities of contamination or release might exist (depending on reactor type and maintenance/intervention).

This is a valid point. Annual total emissions from Beznau were indeed only little smaller in 2015 compared to 2014 despite long maintenance periods. Note that emissions from Beznau were not assumed to be zero after March 2015, because only reactor 1 was shut down, but not reactor 2.

L274: Why is precision of 10-min aggregates reported for JFJ, while long-term reproducibility was reported for Beromunster observations? How are those two quantities combined into one uncertainty for DCO?

Zelleweger et al. 2012 reported a precision of 2.5 ppb and 1.0 ppb for 1-minute and 10-minute averages, respectively for the CRDS CO-analyzer at JFJ. In the uncertainty calculation we have used the 1-minute uncertainty.

L329: Please consider highlighting periods with southeastern European air masses in Figure 2.

As mentioned above we have now included the FLEXPART-COSMO footprints for each sampling period in the supplementary materials, which is more informative. We believe such modification of Figure 2 will not add additional information but rather complicate the figure, as much information is already included.

L349: Please clarify: if the reported uncertainties of RCO (summer and winter) in L347 and L348 are correct there is no significant seasonal difference. Hence, the authors should discuss why there is no difference rather than discuss a reason for a non-existing seasonality.

Despite the fact that the two RCO values are not significantly different considering the uncertainties, the correlation coefficients are different with $r^2$ of only 0.3 in summer (Table 1). Hence, a higher uncertainty was calculated for summer, and comparing with the better-constrained winter values, will be misleading. This is also implicated in the paragraphs following this section where we have only considered the wintertime $R_{CO}$. We have now clarified these points in the manuscript in lines 408-410.

L355: Consider changing to: "The value obtained this way is statistically not different. . ." This could also be discussed more in terms of its implications.

Thank you for your suggestion, we rephrased it in the main text as:

The values obtained in this way (12.7 ± 1.2, $r^2 = 0.6$) is not significantly different from the value obtained using Jungfraujoch as background site. Please note that the JFJ background

represents a clean tropospheric background with a footprint covering a large part of Europe. In the manuscript, we have stated that the $R_{CO}$ value is almost insensitive to the choice of CO background. In both cases, the background values are derived using the REBS smoothing technique.

L371: The authors mention that cold conditions and mass transport from Eastern Europe are likely causes, yet no meteorological data is shown in this paper. Please consider adding a supplement with the key information that you have based this analysis on.

As mentioned above we have now included the model simulations for the air masses during each sampling period in the supplementary materials. We have also added the temperature data from the Beromünster tower at 212.5 m in Figure 2.

L380: The ratios do indeed differ significantly, but you need to establish why Beromünster-JFJ based RCO should be representative for Switzerland (only). See general comment #3

According to a thorough footprint analysis of the Beromünster tower by Oney et al., (2015) (Figure 12) the Beromünster tower foot print is restricted to Switzerland during winter and the $R_{CO}$ values are independent of which background site used while in summer it includes larger areas from neighboring Germany and France. However, during winter/spring air-masses transported from Eastern Europe were reported (Oney et al 2017) and shown from the FLEXPART-COSMO simulations (Supplementary materials).

L407: Colder temperatures are mentioned the main cause, again. If the temperature is such a good predictor of DCO2ff a simple scatter plot should suffice to strengthen your argument.

The temperature record is now additionally included to in Fig. 2.

L425: Please consider adding visual aides to highlight the seasonal cycle in Figure 5. It seems not too clear in the printed version.

Figure 5a shows the all year time series of $CO_{2bio}$ but the data seems noisy due to strong day-night variation in $CO_{2bio}$. However, clear signals are also present for e.g. negative spikes are frequent during summer implying net uptake. Adding a seasonally varying harmonic fit to show seasonal pattern will be misleading considering the strong variability in $CO_{2bio}$ but averaging these values as shown in Figure 6 better illustrates the seasonal variations.

Table 1: An overall of 45 RCO values is reported, while the study is supposedly based on a 3-years long time series (L451) of bi-weekly samples (i.e. 78). Seven samples were accounted for due to the leakage problem reported in L167. What caused the other 26 to be missing here? Were they excluded, not samples, etc?

It is indeed an important point as mentioned by the reviewer that the lower sample amount need clarification. The study period in this manuscript is from 31 July 2013 until 3 December 2015, which is 2.3 years, now corrected in line 508.

During this period, we have collected one sample per month until October 2013 and the biweekly sampling started from December 2013. Hence we have collected 7 samples in 2013, 26 samples in 2014 and 24 samples in 2015 with a total of 57 samples, out of which 7 were excluded due to contamination. From the remaining 50 samples, 5 were excluded due to a strong mismatch among the triplicates in terms of $CO_2$ amount after sample extraction which is indicative of contamination. We have now clarified these points in lines 144 and

182.

Figure 2: Do the dashed lines in 2c and 2d both denote the averages of is the dashed line in 2d just y=0ppm

No the dashed line in Figure 2c indicates the mean $CO_{2ff}$. The one in Fig. 2d is just y=0 ppm, to indicate the sign of the $CO_2$ signal and it is now changed into a solid line.

See comments above

Henne, S., Brunner, D., Oney, B., Leuenberger, M., Eugster, W., Bamberger, I., Meinhardt, F., Steinbacher, M., and Emmenegger, L.: Validation of the Swiss methane emission inventory by atmospheric observations and inverse modelling, Atmos Chem Phys, 16, 3683-3710, 2016.
Oney, B., Henne, S., Gruber, N., Leuenberger, M., Bamberger, I., Eugster, W., and Brunner, D.: The CarboCount CH sites: characterization of a dense greenhouse gas observation network, Atmos. Chem. Phys., 15, 11147-11164, 10.5194/acp-15-11147-2015, 2015.